# Oldest skeleton of a fossil flying squirrel casts new light on the phylogeny of the group

Isaac Casanovas-Vilar[1]*, Joan Garcia-Porta[2], Josep Fortuny[1,3], Óscar Sanisidro[4], Jérôme Prieto[5,6], Marina Querejeta[7], Sergio Llácer[1], Josep M Robles[1], Federico Bernardini[8,9], David M Alba[1]

[1]Institut Català de Paleontologia Miquel Crusafont, Universitat Autònoma de Barcelona, Barcelona, Spain; [2]Centre de Recerca Ecològica i Aplicacions Forestals, Universitat Autònoma de Barcelona, Barcelona, Spain; [3]Centre de Recherches sur les Paléoenvironnements et la Paléobiodiversité, Muséum national d'Histoire naturelle, Paris, France; [4]Biodiversity Institute, University of Kansas, Lawrence, United States; [5]Department für Geo- und Umweltwissenschaften, Paläontologie, Ludwig-Maximilians-Universität München, Munich, Germany; [6]Bayerische Staatssammlung für Paläontologie und Geologie, Munich, Germany; [7]Bavarian State Collection of Zoology, Munich, Germany; [8]Centro Fermi, Museo Storico della Fisica e Centro Studi e Ricerche Enrico Fermi, Roma, Italy; [9]Multidisciplinary Laboratory, The 'Abdus Salam' International Centre for Theoretical Physics, Trieste, Italy

**Abstract** Flying squirrels are the only group of gliding mammals with a remarkable diversity and wide geographical range. However, their evolutionary story is not well known. Thus far, identification of extinct flying squirrels has been exclusively based on dental features, which, contrary to certain postcranial characters, are not unique to them. Therefore, fossils attributed to this clade may indeed belong to other squirrel groups. Here we report the oldest fossil skeleton of a flying squirrel (11.6 Ma) that displays the gliding-related diagnostic features shared by extant forms and allows for a recalibration of the divergence time between tree and flying squirrels. Our phylogenetic analyses combining morphological and molecular data generally support older dates than previous molecular estimates (~23 Ma), being congruent with the inclusion of some of the earliest fossils (~36 Ma) into this clade. They also show that flying squirrels experienced little morphological change for almost 12 million years.
DOI: https://doi.org/10.7554/eLife.39270.001

*For correspondence:
isaac.casanovas@icp.cat

Competing interests: The authors declare that no competing interests exist.

## Introduction

Flying squirrels (Sciurinae, Pteromyini) are the only group of gliding mammals to have achieved a significant diversity (52 species in 15 genera) and wide geographical distribution across Eurasia and North America (*Koprowski et al., 2016*). They have been classically regarded as a distinct subfamily among the Sciuridae (*McKenna and Bell, 1997*; *McLaughlin, 1984*; *Simpson, 1945*), and even sometimes considered a separate family derived from a different group than the remaining sciurids (*De Bruijn and Ünay, 1989*; *Forsyth Major, 1893*; *Mein, 1970*). The fact that presumed fossil flying squirrels are at least as old as (or maybe even older than) the oldest tree squirrels (36.6 – 35.8 Ma) may support the latter hypothesis. However, flying squirrels are currently recognized as a monophyletic clade, as supported by a set of synapomorphies in the wrist (*Thorington, 1984*). The carpal anatomy of flying squirrels is unique, being related to the structures that support the patagium and

**eLife digest** Mammals can walk, hop, swim and fly; a few, like marsupial sugar gliders or colugos, can even glide. With 52 species scattered across the Northern hemisphere, flying squirrels are by far the most successful group that adopted this way of going airborne. To drift from tree to tree, these small animals pack their own 'parachute': a membrane draping between their lower limbs and the long cartilage rods that extend from their wrists. Tiny specialized wrist bones, which are unique to flying squirrels, help to support the cartilaginous extensions.

The origin of flying squirrels is a point of contention: while most genetic studies point towards the group splitting from tree squirrels about 23 million years ago, the oldest remains – mostly cheek teeth – suggest the animals were already soaring through forests 36 million years ago. However, recent studies show that the dental features used to distinguish between gliding and non-gliding squirrels may actually be shared by the two groups.

In 2002, the digging of a dump site in Barcelona unearthed a peculiar skeleton: first a tail and two thigh bones, big enough that the researchers thought it could be the fossil of a small primate. In fact, and much to the disappointment of paleoprimatologists, further excavating revealed that it was a rodent. As the specimen – nearly an entire skeleton – was being prepared, paleontologists insisted that all the 'dirt' attached to the bones had to be carefully screen-washed. From the mud emerged the minuscule specialized wrist bones: the primate-turned-rodent was in fact *Miopetaurista neogrivensis*, an extinct flying squirrel.

Here, Casanovas-Vilar et al. describe the 11.6 million years old fossil, the oldest ever found. The wrist bones reveal that the animal belongs to the group of flying squirrels that have large sizes. Evolutionary analyses that combined molecular and paleontological data demonstrated that flying squirrels evolved from tree squirrels as far back as 31 to 25 million years ago, and possibly even earlier. In addition, the results show that *Miopetaurista* is closely related to *Petaurista*, a modern group of giant flying squirrels. In fact, their skeletons are so similar that the large species that currently inhabit the tropical and subtropical forests of Asia could be considered living fossils.

Molecular and paleontological data are often at odds, but this fossil shows that they can be reconciled and combined to retrace history. Discovering older fossils, or even transitional forms, could help to retrace how flying squirrels took a leap from the rest of their evolutionary tree.
DOI: https://doi.org/10.7554/eLife.39270.002

their particular gliding position, which is different from that of all other gliding mammals (*Thorington, 1984*; *Thorington and Darrow, 2000*). Molecular phylogenies indicate that flying squirrels (tribe Pteromyini) are nested within tree squirrels (subfamily Sciurinae) and likely diverged as recently as the latest Oligocene–early Miocene (23 ± 2.1 Ma) (*Fabre et al., 2012*; *Mercer and Roth, 2003*; *Steppan et al., 2004*). Notwithstanding, the pteromyin fossil record suggests a much older split. Indeed, one of the earliest sciurids, *Hesperopetes thoringtoni* from the late Eocene (36.6 – 35.8 Ma) of North America, has been related to the lineage leading to flying squirrels according to dental morphology (*Emry and Korth, 2007*). In the light of molecular results, it was conceded that *Hesperopetes* unlikely represented a pteromyin and was not assigned to any squirrel subfamily (*Emry and Korth, 2007*). On the other hand, this genus appears to have been closely related to *Oligopetes* (*Emry and Korth, 2007*), an earliest Oligocene (ca. 34 – 31 Ma) purported flying squirrel from Europe and Pakistan (*Cuenca Bescós and Canudo, 1992*; *De Bruijn and Ünay, 1989*; *Heissig, 1979*; *Marivaux and Welcomme, 2003*). *Hesperopetes* is last recorded during the earliest Oligocene (Orellan; *Korth, 2017*), coinciding with the oldest record of *Sciurion* (*Bell, 2004*), yet another alleged flying squirrel. Isolated cheek teeth are the only material available for all these taxa, which have been related to flying squirrels exclusively based on dental morphology. In fact, the whole fossil record of flying squirrels almost exclusively consists of isolated cheek teeth and a few mandibular and maxillary fragments. Unfortunately, dental features commonly used to recognize flying squirrels are not unique but also present in other sciurids (*Thorington et al., 2005*), so it is uncertain if any of the extinct 'flying' squirrels belonged to this group. Furthermore, if any of the oldest (late Eocene–early Oligocene) forms truly represented a pteromyin this would imply a discrepancy of more than 10 Myr between molecular and paleontological data. Contrary to dental material, postcranial

remains do show diagnostic characters of the pteromyins (*Thorington, 1984*; *Thorington et al., 2005*; *Thorington and Darrow, 2000*). Therefore, they are of utmost importance to clarify the assignment of extinct 'flying' squirrels and calibrate their divergence date from other sciurids. Yet, these have not been described and are rarely preserved in the fossil record. Here we report a remarkably complete skeleton of a Miocene squirrel that displays the gliding-related diagnostic features shared by extant pteromyins and allows for a recalibration of the time of origin and diversification of the group. The fossil record of 'flying' squirrels is further discussed in the light of this new finding and the results of our phylogenetic analyses.

## Results

### Recovered material and specific attribution

The described partial skeleton (IPS56468; *Figure 1*, *Videos 1*, 3D model in *Supplementary file 1*) was recovered at Abocador de Can Mata site ACM/C5-D1 (els Hostalets de Pierola, Catalonia, Spain; see Materials and Methods), with an estimated age of 11.63 Ma (*Alba et al., 2017*). The

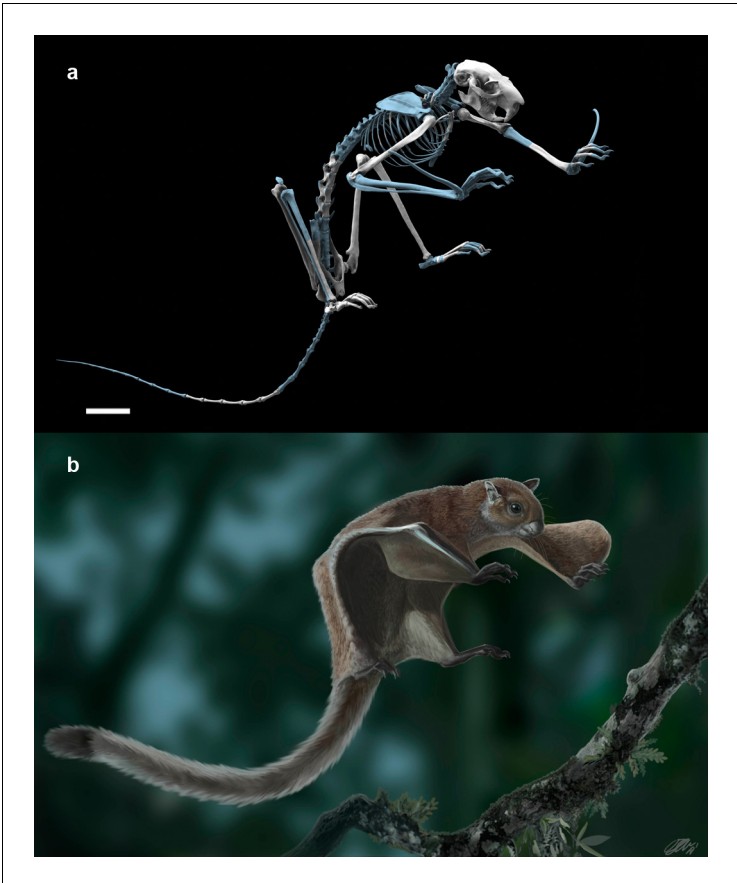

**Figure 1.** The fossil flying squirrel *Miopetaurista neogrivensis*. (a) Reconstruction of the skeleton based in the partial skeleton IPS56468 from Abocador de Can Mata. Missing elements are based on extant giant flying squirrel *Petaurista petaurista* and are colored in blue. (b) Life appearance of *Miopetaurista neogrivensis* showing the animal ready to land on a tree branch. Coat pattern and color are based in extant *Petaurista* species, the sister taxon of *Miopetaurista* (see *Figure 7*). See *Video 1* for an animated version of this reconstruction and 3D model in *Supplementary file 1* to view and manipulate a low-quality model of the skeleton. For recovered elements of the postcranial skeleton see *Figures 2* and *4* and *Table 1*. For a description and comparison of the postcranial bones, see Appendix 3.3. See *Figure 6* and *Video 3* for a more detailed cranial reconstruction. 3D models generated from μCT scan data and photogrammetry. Scale bar is 4 cm.
DOI: https://doi.org/10.7554/eLife.39270.003

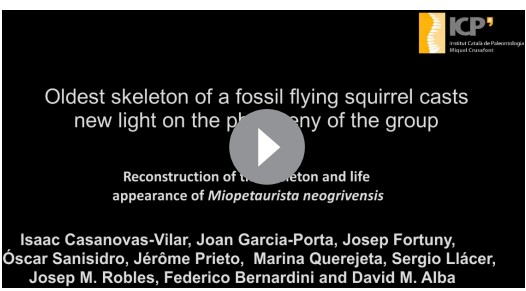

**Video 1.** Reconstruction of the skeleton and life appearance of *Miopetaurista neogrivensis*. The squirrel is shown reducing speed just before landing on a tree branch. Skeleton reconstruction based in the partial skeleton IPS56468 from Abocador de Can Mata. Coat pattern and color are based in extant *Petaurista* species, the sister taxon of *Miopetaurista*. A high-quality 3D surface model of the reconstructed skeleton is available at MorphoBank https://morphobank.org/index.php/Projects/ProjectOverview/project_id/3108
DOI: https://doi.org/10.7554/eLife.39270.012

recovered remains were found partly articulated (*Figure 2—figure supplement 1*) and comprise more than 80 complete and fragmentary bones including the skull (*Figure 6—figure supplement 1*) and elements of the fore- and hindlimbs (*Figures 2–5*, *Figure 2*, *Table 1*). Additional material, including a second cranium (*Figure 6—figure supplement 2*), has been recovered from the same horizon and other roughly coeval ACM localities (*Table 2*). The specimens are assigned to *Miopetaurista neogrivensis* based on diagnostic cheek tooth morphology (*Figure 3*; for detailed description and comparisons of cheek teeth morphology see Appendix 3.1). In the ACM localities a second genus of 'flying' squirrel, *Albanensia*, is recorded, but *Miopetaurista* is clearly distinguished by its larger size, and several morphological features. The diagnostic characters of *M. neogrivensis* comprise: its large size; the presence of a complete entolophid and the frequent occurrence of a short mesolophid in the lower molars; and the large mesostyle in the P4 (*Casanovas-Vilar et al., 2015*; *Mein, 1970*). *Miopetaurista neogrivensis* has only been reported from La Grive L5 (type locality) and L3 in France, from Bellestar (Seu d'Urgell Basin, also in Catalonia), and from several sites from the Vallès-Penedès Basin (*Casanovas-Vilar et al., 2015*). This species is extremely rare, being represented by just a few isolated cheek teeth in most of the Vallès-Penedès sites.

Extant and fossil flying squirrels have been classified into different groups according to the complexity of dental morphology (*Mein, 1970*). The cheek teeth of *Miopetaurista* show a simple occlusal pattern, with enamel wrinkling only in the lower molars and no additional lophules (*Figure 3*, Appendix 3.1). This pattern clearly differs from the more complex one of other large-sized flying squirrels, such as *Aeretes* and *Petaurista* (*Mein, 1970*; *Thorington et al., 2002*). Therefore, *Miopetaurista* has been included within the group that comprises *Aeromys* and the small-sized flying squirrels, which do show simple dental patterns (*Mein, 1970*). However, our phylogenetic analyses (see below) show that *M. neogrivensis* is the sister taxon of extant *Petaurista*, a genus that would belong to a completely different group according to dental classification (*Mein, 1970*). Considering dental morphology *Petaurista* is assigned to a group characterized by its complex dental pattern with additional transverse lophules which would also comprise the genera *Aeretes*, *Belomys*, *Eupetaurus* and *Trogopterus*, among others (*Mein, 1970*). This clearly illustrates that dental characters, although useful to diagnose the different species and genera, should not warrant high consideration for disentangling the phylogenetic relationships between flying squirrels.

## Morphological description and comparisons

Among the recovered postcranial material, a complete scapholunate and the dorsal end of the pisiform (*Figure 4* and *Video 2*) are the most diagnostic elements of pteromyins, because they form the functional complex associated with the extension of the gliding membrane (*Thorington, 1984*; *Thorington et al., 2005*; *Thorington et al., 2002*; *Thorington and Darrow, 2000*). The styliform cartilage, which supports the patagium in all members of the group, attaches to the pisiform and is extended when the wrist is radially abducted and dorsiflexed (*Thorington, 1984*; *Thorington and Darrow, 2000*). The pisiform of *M. neogrivensis* displays an elevated process for the articulation with the scapholunate (*Figure 4* and *Video 2*). This is characteristic of pteromyins, serving as a stabilizer of the styliform cartilage, whereas in other squirrels this bone articulates only with the triquetrum and the distal end of the ulna. Moreover, in the scapholunate of *M. neogrivensis*, the articular surface for the radius is much more convex than in tree squirrels, thus resembling the flying squirrel condition, which enables a greater radial abduction. Therefore, the proximal wrist joint morphology of *M. neogrivensis* indicates that this species belongs to the pteromyin clade and provides the oldest

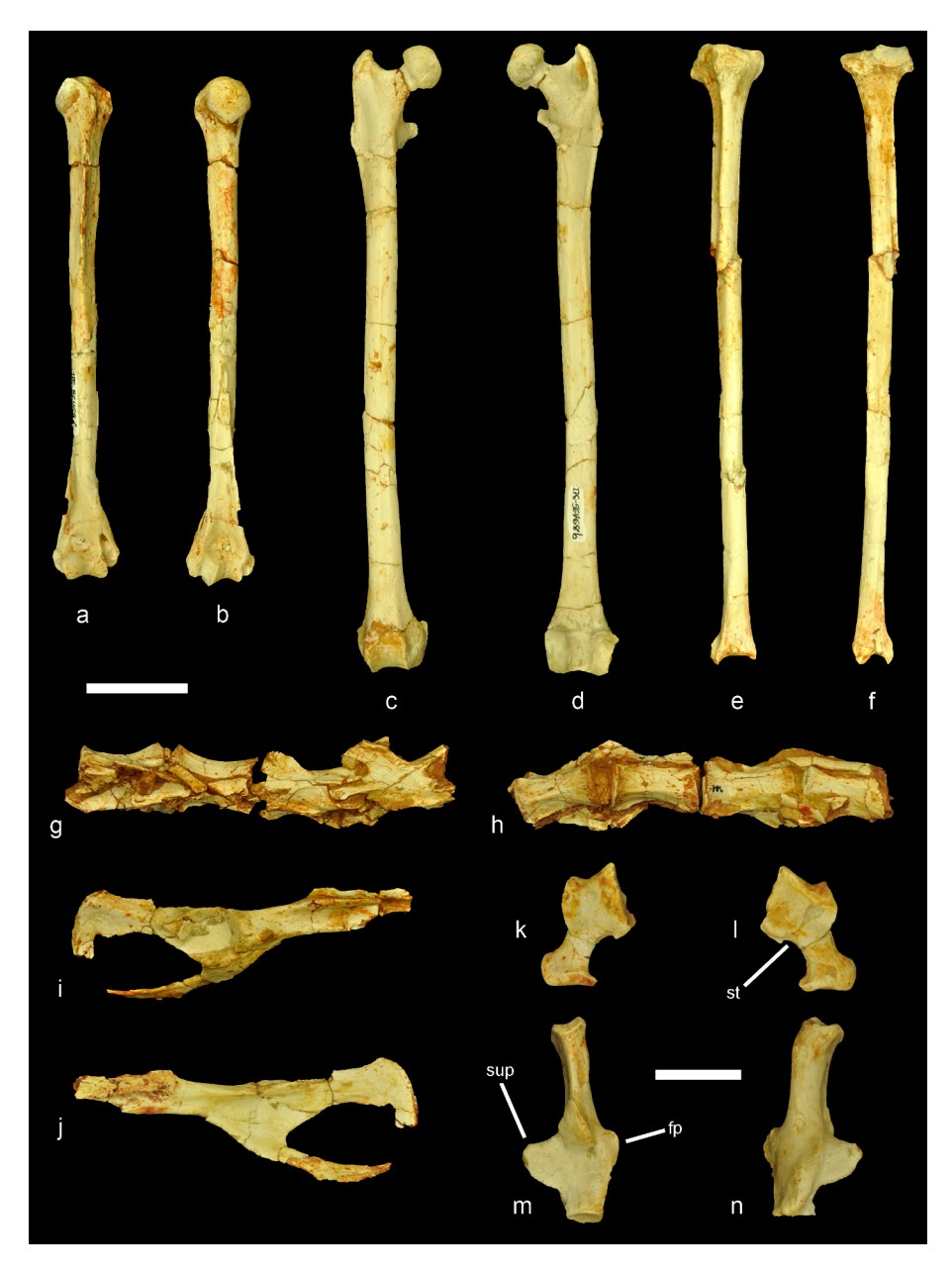

**Figure 2.** Selected postcranial elements of the partial skeleton of *Miopetaurista neogrivensis*. (**a–b**) Right humerus (IPS56468f) in cranial and caudal views. (**c–d**) Right femur (IPS56468b) in cranial and caudal views. (**e–f**) Right tibia (IPS56468a) in cranial and caudal views. (**g–h**) Lumbar vertebrae L3–L6 (IPS56468m–n) in dorsal and ventral views. Note that vertebrae are in anatomical connection. (**i–j**) Partial right coxal (IPS56468k) in lateral and medial views. The proximal end of the ilium is not preserved and part of the pubis is damaged. (**k–l**) Left astragalus (IPS56478t). (**m–n**) Left calcaneus (IPS56468s). fp, fibular process; st, sulcus tali; sup, sustentacular process. Scale bar is 2 cm in figures (**a–j**) and 1 cm in figures (**k–n**). For a reconstruction of the skeleton see *Figure 1*, *Videos 1* and *3D* model in *Supplementary file 1*. Details of particular bones are shown in *Figure 5*; *Figure 2—figure supplement 1*. For a detailed description and comparison of the postcranial bones of *M. neogrivensis* see Appendix 3.3.
DOI: https://doi.org/10.7554/eLife.39270.004

The following figure supplement is available for figure 2:

**Figure supplement 1.** Schematic drawing of the recovered skeletal elements of *Miopetaurista neogrivensis* from Abocador de Can Mata locality ACM/C5-D1 (IPS56468) in the position that they were found.
DOI: https://doi.org/10.7554/eLife.39270.005

**Table 1.** Catalogue of bones and bone fragments composing the partial skeleton of *Miopetaurista neogrivensis*.
This list includes the catalogue numbers (preceded by the acronym 'IPS') of the various bones and bone fragments belonging to the partial skeleton of a single individual of Miopetaurista neogrivensis (IPS56468) from locality ACM/C5-D1. IPS, acronym for the collections of the Institut Català de Paleontologia Miquel Crusafont.

| Catalogue no. | Region | Description |
|---|---|---|
| IPS56468a | leg | right tibia |
| IPS56468b | leg | right femur |
| IPS56468c | leg | left femur |
| IPS56468d | leg | distal half of the left tibia |
| IPS56468e | arm | left humerus |
| IPS56468f | arm | right humerus |
| IPS56468g | arm | distal half of the right radius, with damaged epiphysis |
| IPS56468h | cranium | almost complete cranium, laterally compressed in an oblique angle |
| IPS56468i | cranium + tail | partial right mandible (angular process broken), caudal vertebra (probably corresponding to the mid part of the tail) |
| IPS56468j | cranium | partial left mandible (articular process broken, all other processes with minor damage) |
| IPS56468k | pelvic girdle | partial right coxal (proximal end of the ilium and part of the pubis damaged) |
| IPS56468l | pelvic girdle | partial left coxal (missing most of the pubis and ischium, extensive damage in the ilium) |
| IPS56468m | trunk | lumbar vertebrae (L3–L4) in anatomical connection |
| IPS56468n | trunk | lumbar vertebrae (L5–L6) in anatomical connection |
| IPS56468o | neck | partial axis (only part of the vertebral body is preserved) and partial cervical vertebra (C3) |
| IPS56468p | trunk | thoracic vertebra (T1?) |
| IPS56468q | tail | four caudal vertebrae (mid part of the tail) that articulate with one another |
| IPS56468r | trunk | seven rib fragments |
| IPS56468s | ankle | left calcaneus and left navicular |
| IPS56468t | ankle | left astragalus |
| IPS56468u | trunk | three partial thoracic vertebrae (T2–T4?) that articulate with one another |
| IPS56468v | indeterminate | associated bone fragments (may not belong to *M. neogrivensis*) |
| IPS56468w | indeterminate | associated bone fragments (may not belong to *M. neogrivensis*) |
| IPS56468x | foot? | six distal phalanges which are not assigned to any particular ray or side; attribution to the foot is tentative |
| IPS56468y | trunk | two sternebrae that articulate with one another |
| IPS56468z | foot | left metatarsals 2–4 in anatomical connection |
| IPS56468aa | thoracic girdle | right clavicle (with minor damage in its acromial end) and partial left clavicle (acromial end missing) |
| IPS56468ab | foot | complete right metatarsal 3, and partial rigth metatarsals 4 (proximal end missing), 2 and 4 (only distal half preserved) |
| IPS56468ac | foot | four proximal phalanges and one partial proximal phalanx (distal half); they are not assigned to any particular ray or side |
| IPS56468ad | foot? | seven intermediate phalanges and five fragments; they are not assigned to any particular ray or side and attribution to the foot is tentative |
| IPS56468ae | hand | four proximal phalanges; they are not assigned to any particular ray or side |
| IPS56468af | ankle | left intermediate and medial cuneiform |
| IPS56468ag | ankle | right navicular |
| IPS56468ah | wrist | right scapholunate and dorsal end of the right pisiform |

DOI: https://doi.org/10.7554/eLife.39270.010

evidence of gliding locomotion in sciurids (see also Appendix 3.3). The latter is further confirmed by other postcranial adaptations shared with extant pteromyins (*Thorington et al., 2005*), including the elongated and slender limb bones with reduced muscular attachments (*Figure 5* and *Figure 5—figure supplement 1*), which enhance joint extension during gliding (*Thorington et al., 2005*), as well

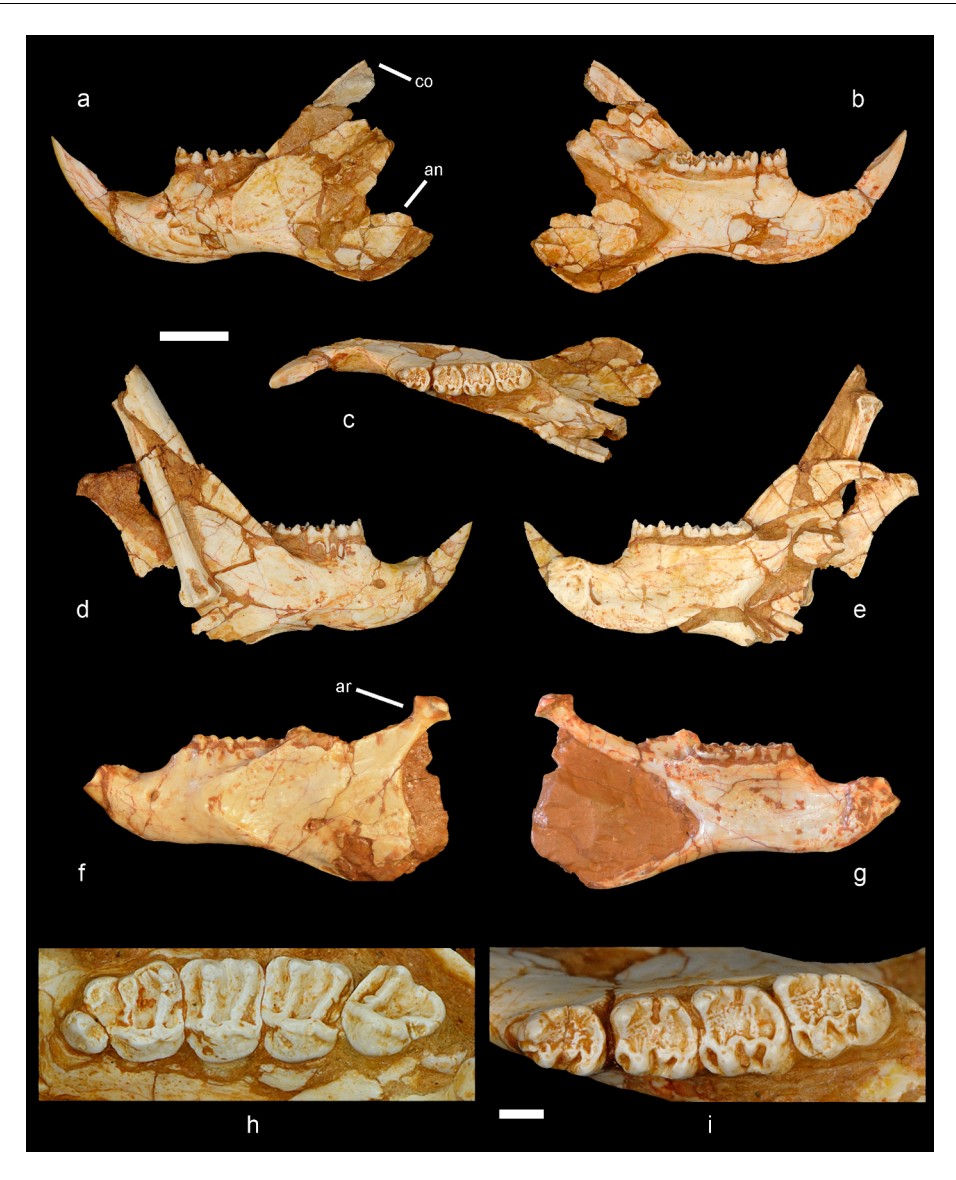

**Figure 3.** Mandible and cheek teeth of *Miopetaurista neogrivensis*. (a to c) Partial left hemimandible (IPS56468j) in lateral, medial and dorsal views. (d to e) Partial right hemimandible (IPS56468i) in lateral and medial views. A caudal vertebra and a bone fragment are attached to the lateral side of the mandibular ramus. Both hemimandibles were associated to the partial skeleton IPS56468 from ACM/C5-D1. (f to g) Partial hemimandible (IPS87560) from ACM/C8-B sector in lateral and medial views. (h) Left upper cheek teeth series (P3–M3) of IPS56468h (*Figure 6—Figure supplement 1* ). (i) Left lower cheek teeth series (p4–m3) of IPS56468j. Cheek teeth measurements are given in *Supplementary file 4* whereas mandibular measurements are given in *Supplementary file 6*. For a detailed description and comparisons of cheek teeth and mandible morphology see Appendix 3.1 and 3.2. an, angular process; ar, articular process; co, coronoid process. Scale bar is 1 cm in figs. a to g; 2 mm in (h to i).
DOI: https://doi.org/10.7554/eLife.39270.006

as the elongated lumbar vertebrae (*Figure 2*; see Appendix 3.3 for a detailed description and comparisons of the postcranial elements). The elongation of lumbar vertebrae and limbs determines the size and shape of the patagium and dictates important aerodynamic features, such as the decreased wing loading of flying squirrels (*Thorington and Heaney, 1981*; *Thorington and Santana, 2007*).

Based on morphological (*Thorington et al., 2002*) and molecular data (*Mercer and Roth, 2003*), flying squirrels are divided into two distinct subtribes: Pteromyina, comprising large-sized forms, and Glaucomyina, for the small-sized ones (*Thorington and Hoffmann, 2005*). The skeleton of *M. neogrivensis* morphologically resembles that of the Pteromyina, further being comparable in size to their

**Table 2.** Catalogue of additional material of *Miopetaurista neogrivensis*.

This list includes the additional material of *Miopetaurista neogrivensis* from locality ACM/C5-D1 and the approximately stratigraphicaly equivalent localities ACM/C8-Af and ACM/C6-A5, as well as from sector ACM/C8-B. IPS, acronym for the collections of the Institut Català de Paleontologia Miquel Crusafont.

| Catalogue no. | Locality | Anatomical element |
|---|---|---|
| IPS43480 | ACM/C5-D1 | L m1 |
| IPS43505 | ACM/C5-D1 | L m3 |
| IPS43675 | ACM/C5-D1 | R P4 |
| IPS43677 | ACM/C5-D1 | partial left mandible with p4–m3 (only part of the mandibular body preserved) |
| IPS43724 | ACM/C5-D1 | right maxillary fragment with partial P4-M2 |
| IPS77856 | ACM/C5-D1 | R P4 |
| IPS77857 | ACM/C5-D1 | R P4 |
| IPS77858 | ACM/C5-D1 | R M1/M2 |
| IPS77859 | ACM/C5-D1 | R M1/M2 |
| IPS77860 | ACM/C5-D1 | fragment of R M1/M2 |
| IPS77861 | ACM/C5-D1 | fragment of R M1/M2 |
| IPS77862 | ACM/C5-D1 | L M1/M2 |
| IPS77863 | ACM/C5-D1 | broken L M1/M2 |
| IPS77864 | ACM/C5-D1 | fragment of L M1/M2 |
| IPS77865 | ACM/C5-D1 | fragment of L M1/M2 |
| IPS77866 | ACM/C5-D1 | L M3 |
| IPS77867 | ACM/C5-D1 | R M3 |
| IPS77868 | ACM/C5-D1 | L dp4 |
| IPS77869 | ACM/C5-D1 | fragment of L dp4 |
| IPS77870 | ACM/C5-D1 | R dp4 |
| IPS77871 | ACM/C5-D1 | L p4 |
| IPS77872 | ACM/C5-D1 | R p4 |
| IPS77873 | ACM/C5-D1 | R p4 |
| IPS77874 | ACM/C5-D1 | L m1 |
| IPS77875 | ACM/C5-D1 | L m1 |
| IPS77876 | ACM/C5-D1 | L m1 |
| IPS77877 | ACM/C5-D1 | L m2 |
| IPS77878 | ACM/C5-D1 | L m2 |
| IPS77879 | ACM/C5-D1 | broken L m1/m2 |
| IPS77880 | ACM/C5-D1 | abraded L m1/m2 |
| IPS77881 | ACM/C5-D1 | fragment of L m1/m2 |
| IPS77882 | ACM/C5-D1 | fragment of L m1/m2 |
| IPS77883 | ACM/C5-D1 | R m1 |
| IPS77884 | ACM/C5-D1 | lingually abraded R m1 |
| IPS77885 | ACM/C5-D1 | broken R m1/m2 |
| IPS77886 | ACM/C5-D1 | R m3 |
| IPS77887 | ACM/C5-D1 | fragment of upper molar |
| IPS77888 | ACM/C5-D1 | fragment of lower molar |
| IPS78179 | ACM/C5-D1 | L M1/M2 |
| IPS85340 | ACM/C5-D1 | fragment of left mandible with m1–m3 and associated p4 |

*Table 2 continued on next page*

*Table 2 continued*

| Catalogue no. | Locality | Anatomical element |
|---|---|---|
| IPS85410 | ACM/C6-A5 | partial cranium (includes the dorsal half of the skull as well as part of the right zygomatic arch and maxillary bone with damaged P4–M3) |
| IPS87560 | ACM/C8-B sector | partial left mandible with p4–m3 (incisor and coronoid process broken, p4 damaged) |
| IPS88677 | ACM/C8-Af | partial cranium (dorsoventrally crushed) |

DOI: https://doi.org/10.7554/eLife.39270.011

largest representatives. Body mass was estimated by means of an allometric regression of body mass vs. skull length in extant sciurids (see Materials and Methods), resulting in 1339 g (50% confidence intervals 1116 – 1606 g thus being in the range of most species of the extant giant flying squirrel *Petaurista* (about 1200 – 2000 g; *Thorington et al., 2012*). The long bones are almost indistinguishable of *Petaurista*. The skull, which was virtually reconstructed from two well-preserved specimens (*Figure 6*, *Figure 6—figure supplement 1-2*, *Video 3*, *Table 3*), is strikingly similar in

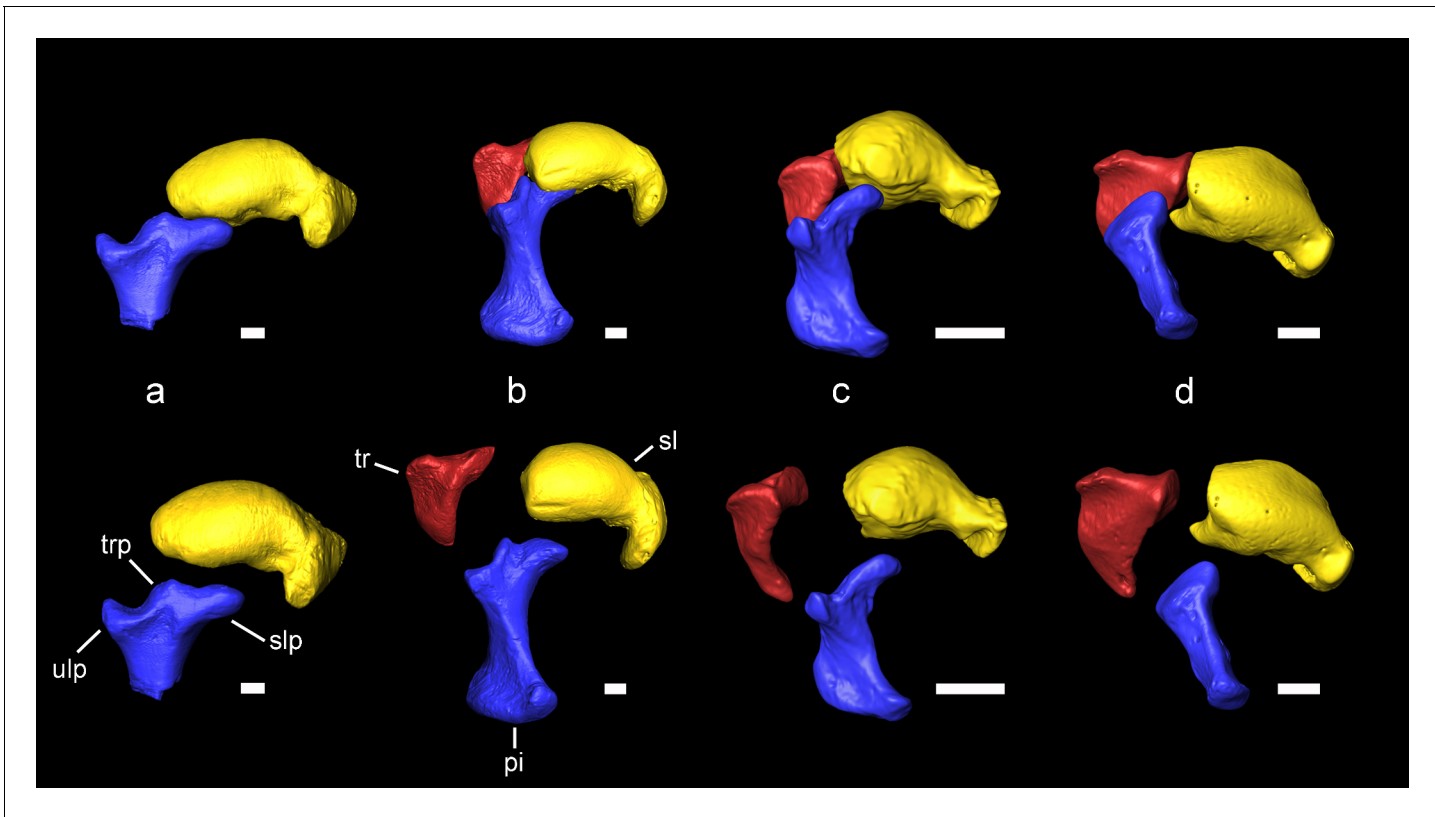

**Figure 4.** Carpal bones associated with the extension of the patagium of *Miopetaurista neogrivensis* as compared to extant squirrels. Articulated bones are shown on top and disarticulated ones are shown below. (a) *Miopetaurista neogrivensis*. (b) *Petaurista petaurista*, large-sized flying squirrel, subtribe Pteromyina. (c) *Hylopetes sagitta*, small-sized flying squirrel, subtribe Glaucomyina. (d) *Sciurus vulgaris*, tree squirrel, tribe Sciurini. The patagium is supported by the styliform cartilage which is attached to the pisiform bone. Flying squirrels present an elevated process for articulation with the scapholunate in the pisiform, whereas in tree squirrels this bone only articulates with the triquetrum and the ulna. In addition, note the presence of a triquetral process in *Miopetaurista* and *Petaurista*, characteristic of the Pteromyina. All extant specimens are kept in the collections of the Naturalis Biodiversity Center (Leiden, the Netherlands). See *Video 2* for an animated version of this figure. Collection numbers of the scanned specimens and computed tomography parameters used are given in *Table 5*. Tridimensional models generated from μCT scan data. pi, pisiform (in blue); sl, scapholunate (in yellow); slp, scapholunate process of the pisiform; tr, triquetrum (in red); trp, triquetral processes of the pisiform; ulp, ulnar process of the pisiform. Scale bar is 1 cm.

DOI: https://doi.org/10.7554/eLife.39270.007

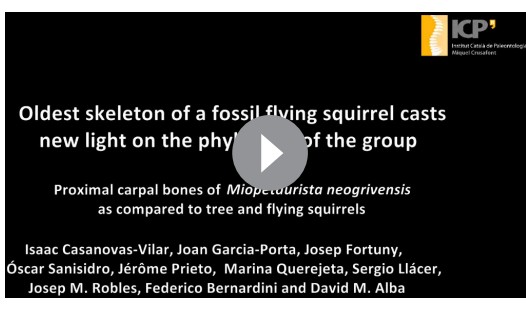

**Video 2.** Proximal carpal bones of *Miopetaurista neogrivensis* as compared to tree and flying squirrels. *Miopetaurista neogrivensis* is compared to *Petaurista petaurista* (large-sized flying squirrel; subtribe Pteromyina); *Hylopetes sagitta* (small-sized flying squirrel; subtribe Glaucomyina); and *Sciurus vulgaris* (tree squirrel; tribe Sciurini). These bones form the morphofunctional complex associated with extension of the patagium. The flying membrane is supported by the styliform cartilage which is attached to the pisiform bone. High-quality 3D surface models of the carpal bones of *Miopeataurista* and other squirrels are available at MorphoBank https://morphobank.org/index.php/Projects/ProjectOverview/project_id/3108
DOI: https://doi.org/10.7554/eLife.39270.013

size and morphology to that of the other large-sized flying squirrels, particularly *Aeromys* and *Petaurista* (for a detailed morphological description of the skull and comparisons see Appendix 3.2). These genera are characterized by their short and wide rostrum, moderately inflated bullae and relatively wide posterior region of the skull. Other morphological similarities include the robust and long postorbital process that partially encloses the orbit, the well-developed jugal process in the zygomatic arch and the presence of two septa in the tympanic cavity (*Video 4*; Appendix 3.2). Most of the smaller flying squirrels show more elongate muzzles, slender or shorter postorbital processes and, in some cases, a higher number of transbullar septa. The proximal carpal bones of *M. neogrivensis* not only unambiguously indicate that it is a flying squirrel, but also allow assigning it to the Pteromyina (*Figure 4* and *Video 2*). The pisiform displays a distinct spur (triquetral process) that fits between the palmar surfaces of the scapholunate and the triquetrum. This process is completely lacking in the Glaucomyina (*Thorington et al., 2002*; *Thorington and Darrow, 2000*) (*Figure 4* and *Video 2*; see also Appendix 3.3). Both subtribes are also distinguished by the origin of the tibio-carpalis muscle, which runs from the ankle to the tip of the styliform cartilage, defining the edge of the patagium. In the Glaucomyina the tibiocarpalis originates from a tuberosity on the distal tibia which is lacking in *M. neogrivensis* and the Pteromyina (*Thorington et al., 2002*). In the latter, the tibiocarpalis originates from the metatarsals instead.

## Phylogenetic results

The assignment of *M. neogrivensis* to the subtribe Pteromyina is further confirmed by a total evidence phylogenetic analysis combining morphological and molecular data (see Materials and Methods). This analysis relied on 35 extant species of sciurids plus *Aplodontia* used as outgroup (both with molecular and morphological data) and two fossils (*M. neogrivensis* and the oldest-known tree squirrel, *Douglassciurus jeffersoni*, both represented only by morphological data; see also Appendices 1.1 and 2). Our analysis strongly supports *M. neogrivensis* as the sister taxon of the *Petaurista* spp. clade and indicates a divergence date between flying and tree squirrels ranging from the late Eocene to the late Oligocene (95% highest posterior density [HPD] interval 36.5 – 24.9 Ma; *Figure 7*). Such range is congruent with previous molecular estimates (*Fabre et al., 2012*; *Mercer and Roth, 2003*) but also supports older dates, as old as the oldest records of purported pteromyins (*Emry and Korth, 2007*). The Pteromyina and Glaucomyina would have diverged between the late Oligocene–early Miocene (HPD interval 27.1 – 18.1 Ma; *Figure 7*). We independently tested the estimates of Pteromyini/Sciurini divergence and the onset of Pteromyini crown diversification by means of a node dating analysis (see Materials and Methods) of the extant Sciurinae using two different calibration points, one for each tribe (*Figure 7—figure supplement 1*, *Table 4*; see also Appendix 1.2). Estimates for many nodes are somewhat younger using this alternative approach, but mostly overlap with the younger half of HPD intervals of the total evidence analysis (*Figure 7*). The age range for the Sciurini/Pteromyini split spans from the mid Oligocene to early Miocene (HPD interval 30.6 – 17.4 Ma; *Figure 7—figure supplement 1*). Therefore, divergence estimates derived using independent methods overlap for the late Oligocene.

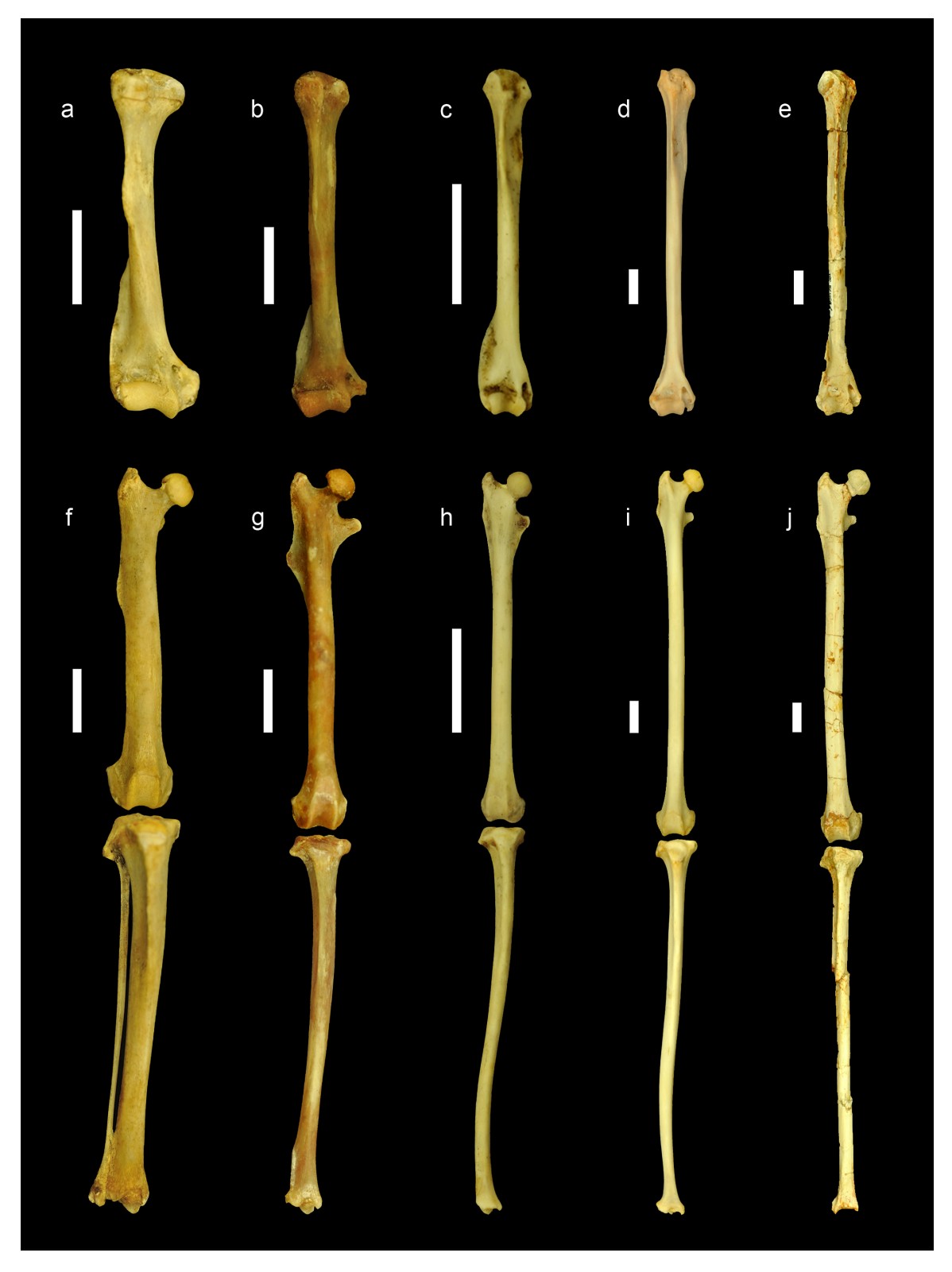

**Figure 5.** Comparison of the limb bones of extant ground, tree and flying squirrels with *Miopetaurista neogrivensis*. All elements are scaled to femur length and shown in anterior view. Humerus (**a–e**) and articulated femur and tibia (**f–j**) of: (**a,f**) the xerin ground squirrel *Xerus erythropus*; (**b,g**) the callosciurin tree squirrel *Callosciurus prevostii*; (**c,h**) the small-sized flying squirrel (subtribe Glaucomyina) *Hylopetes sagitta*; (**d,i**) the large-sized fying squirrel (subtribe Pteromyina) *Petaurista petaurista*; (**e,j**) *Miopetaurista neogrivensis.* Note that limb bones of flying squirrels and *M. neogrivensis* are

*Figure 5 continued on next page*

*Figure 5 continued*

much longer and more slender than those of tree and ground squirrels. Furthermore, processes and areas for the insertion of the main limb muscles are reduced. For a description and comparison of the postcranial bones of *M. neogrivensis*, see Appendix 3.3. See **Supplementary file 7** for the collection numbers of the figured specimens and postcranial measurements. All bones are right elements, except for a–b and f–g, which are reversed left elements. Scale bar is 1 cm.

DOI: https://doi.org/10.7554/eLife.39270.008

The following figure supplement is available for figure 5:

**Figure supplement 1.** Comparison of the proximal and distal ends of the humerus, femur and tibia of extant ground, tree and flying squirrels with *Miopetaurista neogrivensis*.

DOI: https://doi.org/10.7554/eLife.39270.009

## Discussion

### Divergence date between flying and tree squirrels

The two independent phylogenetic analyses estimate divergence dates ranging from 36.5 to 17.4 Ma (late Eocene–early Miocene; *Figure 7* and *Figure 7—figure supplement 1*). These broad ranges are congruent with a previous molecular phylogenetic analysis of the sciurids (*Mercer and Roth, 2003*), which placed the split between tree and flying squirrels near the Oligocene/Miocene boundary (23 ± 2.1 Ma), although allowing for substantially older dates. Total evidence analysis in particular provides older estimates (36.5 – 24.9 Ma) than node dating (30.6 – 17.4 Ma), but these still marginally overlap with those of previous molecular results. The different results may arise from the different selection of calibration points (see *Table 4*). In that study (*Mercer and Roth, 2003*) the age of *Douglassciurus* (ca. 36 Ma) was assigned to the base of the sciurid crown radiation (i.e. the origin of extant major clades) to calibrate the phylogenetic tree whereas the root of our phylogenetic trees is calibrated using multiple points (see Materials and Methods).

A few molecular phylogenetic studies (*Montgelard et al., 2008*; *Tapaltsyan et al., 2015*), generally dealing with the whole rodent order and including only a few sciurid genera, have provided older ages for the divergence of flying squirrels dating back to the late Eocene and earliest Oligocene (34.5 – 30.9 Ma). Again, this may be attributed to the different selection of calibration points, which in most cases consider paleontological data from other rodent groups. Finally, rodent diversification has been analyzed using a molecular supermatrix that included 98% of extant squirrel genera (*Fabre et al., 2012*). This study found a late Oligocene divergence date between pteromyins and sciurins, which is perfectly congruent with our results.

Total evidence 95% highest posterior density (HPD) interval for the pteromyin divergence is very broad (36.5–24.9 Ma; *Figure 7*), being consistent with the range of *Oligopetes* (*Cuenca Bescós and Canudo, 1992*; *De Bruijn and Ünay, 1989*; *Heissig, 1979*; *Marivaux and Welcomme, 2003*), the oldest records of *Sciurion* (*Bell, 2004*) and even the earliest (late Eocene) occurrences of *Hesperopetes* (*Korth, 2017*) (*Figure 8*). In contrast, node dating analysis estimates a younger HPD interval that would exclude the older records of these genera (30.6 – 17.4 Ma; *Figure 7—figure supplement 1*). According to our results, late Eocene to early Oligocene alleged flying squirrels might indeed belong to this group, but this is only supported by total evidence analysis (*Figure 7*). Therefore, our analyses are not conclusive to this regard and further fossil data are required to elucidate the phylogenetic position of these older 'flying' squirrels. Although allowing for older and younger ages, the independently derived estimates are generally older than previous molecular results (*Mercer and Roth, 2003*) and overlap for the late Oligocene, which should be considered the most likely time for pteromyin divergence.

### Flying squirrel radiation and biogeography in relation to Miocene climatic changes

Except for the partial skeleton of *Miopetaurista neogrivensis* described here, no diagnostic postcranial material has been recovered for any other extinct 'flying' squirrel, so their assignment to the pteromyins is doubtful. Here we discuss the fossil record of purported pteromyins in the light of our phylogenetic results, but in these instances our conclusions depend of the correctness of the tribe assignment.

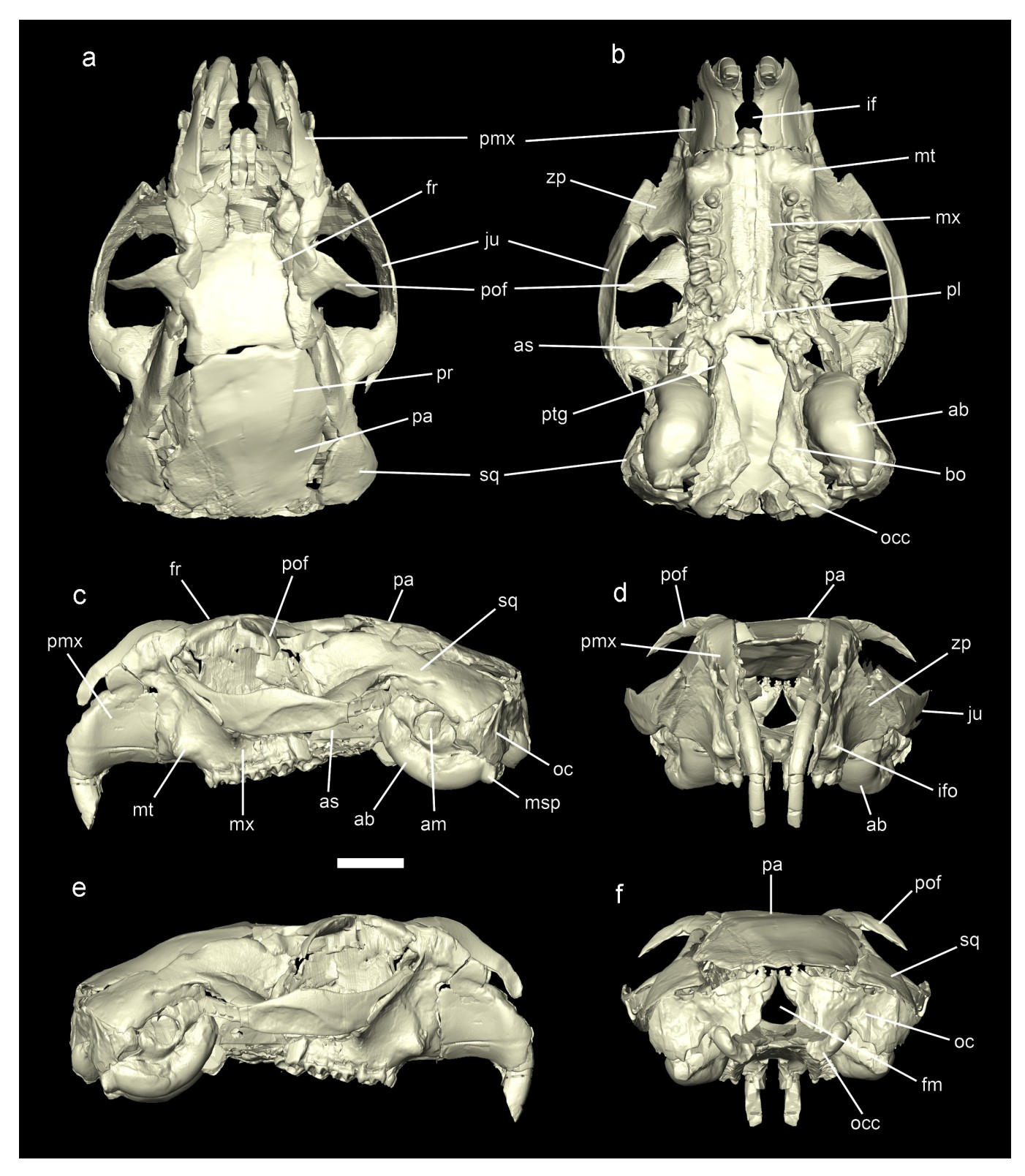

**Figure 6.** Reconstruction of the cranium of *Miopetaurista neogrivensis*. Virtual reconstruction based on μCT scan data from specimens IPS56468h (see *Figure 6—figure supplement 1*) and IPS88677 (see *Figure 6—figure supplement 2*). Specimen IPS56468h was used as the basis for the reconstruction, with missing elements taken from IPS88677 (*Table 3*. (a) Dorsal view. (b) Ventral view. (c) Lateral (left) view. (d) Anterior view. (e) Lateral (right) view. (f) Posterior view. See *Video 3* for an animated version of the skull reconstruction. For a detailed description of skull morphology see

*Figure 6 continued on next page*

*Figure 6 continued*

Appendix 3.2. Cranial measurements for original fossil specimens (IPS56468h, IPS88677) as well as for the virtually reconstructed cranium are given in **Supplementary file 5**. ab, auditory bulla; am, auditory meatus; as, alisphenoid; bo, basioccipital; fm, foramen magnum; fr, frontal; if, incisive foramen; ifo, infraorbital foramen; ju, jugal; msp, mastoid process; mt, masseteric tubercle; mx, maxillary; oc, occipital; occ, occipital condyle; pa, parietal; pl, palatine; pmx, premaximallary; pof, postorbital process of the frontal; pr, parietal ridges; ptg, pterygoid; sq, squamosal; zp, zygomatic plate. Scale bar is 1 cm.

DOI: https://doi.org/10.7554/eLife.39270.014

The following figure supplements are available for figure 6:

**Figure supplement 1.** Cranium of *Miopetaurista neogrivensis* associated with partial skeleton IPS56468.

DOI: https://doi.org/10.7554/eLife.39270.015

**Figure supplement 2** Cranium of *Miopetaurista neogrivensis* (IPS88677) from Abocador de Can Mata locality ACM/C8-Af.

DOI: https://doi.org/10.7554/eLife.39270.016

Our analyses indicate that the initial diversification of flying squirrels into the two extant subtribes (Glaucomyina and Pteromyina) occurred between the latest Oligocene and the middle Miocene, with estimates derived from both methods overlapping for the early Miocene (**Figure 7** and **Figure 7—figure supplement 1**). Such time interval coincides with the almost simultaneous earliest records of new genera of 'flying' squirrels in both North America and Europe. In Europe, these include the small-sized genus *Blackia*, which is first recorded near the Oligocene/Miocene transition (biozone MP30) (**Engesser and Storch, 2008**) at around 23.3 – 23.0 Ma (**Figure 8**). According to dental morphology this genus appears to be closely related to the older North American *Sciurion* (**Skwara, 1986**), so in case their pteromyin affinities were confirmed this would argue for a North American origin of the group. Besides *Blackia* three additional genera are recorded in the European early Miocene (**De Bruijn, 1999**): *Aliveria*, *Miopetaurista* and *Neopetes* (**Figure 8**). These occurrences date back to biozone MN3, corresponding to the early Miocene, yielding a relative age of 20.0 – 18.0 Ma. A species of the extant genus *Hylopetes* has been erected based on material from site of Oberdorf (Austria), correlated to the MN4 (18.0 – 17.0 Ma) (**De Bruijn, 1998**). Such occurrence would pull back the range of this genus into the early Miocene and has been taken for granted in some recent molecular studies (**Lu et al., 2013**) even though it blatantly disagrees with previous molecular results (**Mercer and Roth, 2003**). Indeed, some paleontologists have argued that the characters justifying the ascription of the Austrian and other material to *Hylopetes* are symplesiomorphies shared with many other flying squirrel genera (e.g. *Petinomys*, *Glaucomys*) and assign this material to the extinct 'flying' squirrel genera *Neopetes* and *Pliopetes* (**Casanovas-Vilar et al., 2015**; **Daxner-Höck, 2004**). In North America, aside from *Sciurion* the larger-sized genus *Petauristodon* is also present from the late Oligocene to the late Miocene (**Goodwin, 2008**) (**Figure 8**). While *Sciurion* is already known since the early Oligocene (33.5-33.0 Ma) (**Bell, 2004**), the oldest record of *Petauristodon* is a single molar dated between 25.9 and 23.8 Ma from the John Day Formation of Oregon (**Korth and Samuels, 2015**). Subsequent records date back to the early Miocene (ca. 19 Ma; **Goodwin, 2008**). Interestingly, the specimens ascribed to *Sciurion* and *Petauristodon* were previously referred to the European genera *Blackia* and *Miopetaurista*, respectively (**Goodwin, 2008**). Finally, in Asia the genera *Meinia* (**Qiu, 1981**), *Parapetaurista* (**Qiu, 1981**) and *Shuanggouia* (**Qiu and Liu, 1986**) have been reported from the early Miocene (Shanwangian) Shanwang and Xiacaowan formations of North and Eastern China, with an estimated age of ca. 18 – 16 Ma or slightly younger (**Qiu et al.,**

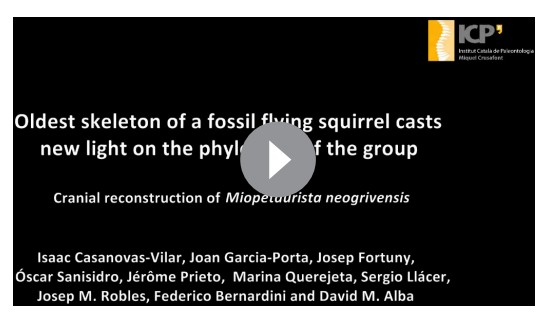

**Video 3.** Cranial reconstruction of *Miopetaurista neogrivensis*. Virtual reconstruction based on µCT scan data from specimens IPS56468h and IPS88677. Specimen IPS56468h was used as the basis for the reconstruction, with missing elements taken from IPS88677. A high-quality 3D surface model of the reconstructed cranium is available at MorphoBank https://morphobank.org/index.php/Projects/ProjectOverview/project_id/3108.

DOI: https://doi.org/10.7554/eLife.39270.017

**Table 3.** Cranial reconstruction of *Miopetaurista neogrivensis*.
List of cranial elements used in the reconstruction of the complete skull indicating catalog no. of each bone or bone fragment and, when appropriate, which side (left or right) was mirrored.

| Cranial element | Side | Catalog. no |
| --- | --- | --- |
| alisphenoid | left | IPS88677 |
| auditory bulla | left | IPS56468h |
| basisphenoid | left | IPS56468h |
| basioccipital and occipital condyle | left | IPS56468h |
| cheek teeth | right | IPS56468h |
| frontal | – | IPS56468h |
| frontal (orbital part) | left | IPS88677 |
| hamular process of pterygoid | right | IPS56468h |
| incisor | left | IPS56468h |
| jugal | left | IPS88677 |
| jugal (posterior half) | right | IPS88677 |
| maxillary | right | IPS56468h |
| occipital (right half) | right | IPS88677 |
| palatines | – | IPS88677 |
| parietal | – | IPS56468h |
| postorbital process of frontal | right | IPS88677 |
| pterygoid | left | IPS56468h |
| pterygoid fossa (alisphenoid) | – | IPS56468h, IPS88677 |
| premaxillary | right | IPS56468h |
| premaxillary (dorsal fragments) | right, left | IPS88677 |
| squamosal | left | IPS56468h |
| squamosal (zygomatic process) | right | IPS88677 |
| zygomatic plate (maxillary) | left | IPS88677 |

DOI: https://doi.org/10.7554/eLife.39270.019

*2013b*) (*Figure 8*). Even though fossil evidence can only justify the ascription of *Miopetaurista* to the pteromyins, it is worth noting that Europe records highest 'flying' squirrel diversity at the time (*Figure 8*), which has led to the suggestion that the group may well have originated there and immediately dispersed into Asia and North America near the Oligocene/Miocene transition (*Lu et al., 2013*). Notwithstanding, this hypothesis is challenged by the fossil record of 'flying' squirrels since the oldest occurrences are in North America. If certain Oligocene forms can be ultimately assigned to the pteromyins this would confirm an opposed model with an early origin and initial diversification in North America and a later dispersal into Eurasia by the latest Oligocene.

Initial flying squirrel diversification and dispersal across the Northern Hemisphere coincided with a period of high mean global temperatures during the early Miocene that peaked between 17 to 15 Ma, during the so-called Mid-Miocene Climatic Optimum (*Zachos et al., 2001*) (*Figure 8*). Humid warm-temperate broadleaf and mixed forests, resembling those existing in the southeastern coast of Asia, characterized the mid latitudes in Eurasia and North America (*Pound et al., 2012*). These forests provided a suitable habitat for flying squirrels and would have contributed to their initial radiation and dispersal into different continents. Our phylogenetic analyses show that some extant genera, such as *Petaurista*, *Pteromys* and *Glaucomys*, diverged approximately at that time (*Figure 7* and *Figure 7—figure supplement 1*), thus agreeing with previous molecular results (*Arbogast et al., 2017*; *Mercer and Roth, 2003*).

The warm early Miocene phase was followed by a gradual cooling and the reestablishment of permanent major ice sheets on Antarctica by about 10 Ma (*Zachos et al., 2001*). Warm-temperate

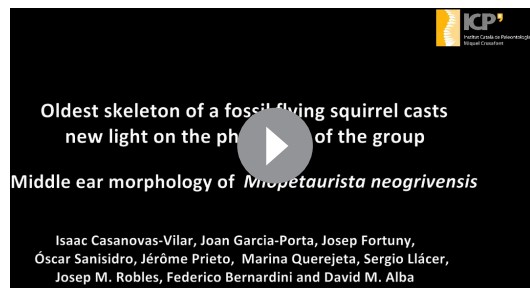

**Video 4.** Middle ear anatomy of *Miopetaurista neogrivensis* as compared to *Petaurista petaurista*. Middle ear cavity anatomy in *Miopetaurista neogrivensis* (left; IPS88677) and the extant giant flying squirrel *Petaurista petaurista* (right; ZMA13418) reconstructed from μCT scan data. Note the presence of two ventral transbullar septa. Most flying squirrels show just two septa except, for a few Glaucomyina (*Petinomys*, *Trogopterus*, *Pteromyscus*) that exhibit a more complex pattern (see Appendix 3.2). Scale bar is 1 cm.

DOI: https://doi.org/10.7554/eLife.39270.018

forests were still dominant in the mid latitudes during the middle and the beginning of the late Miocene. However, their distribution became more restricted, particularly in Western North America and Central Asia, where they began to be replaced by cooler and drier biomes (*Pound et al., 2012*). 'Flying' squirrels are particularly diverse in Europe during this interval, with as many as five different species co-occurring in a single site (*Casanovas-Vilar et al., 2015*). In addition, certain genera such as *Miopetaurista*, *Pliopetaurista* and *Albanensia*, are widely distributed (*Casanovas-Vilar et al., 2015*) (*Figure 8*). *Miopetaurista* and *Pliopetaurista* are also know from the beginning of the late Miocene of Amuwusu (Inner Mongolia, China) (*Qiu et al., 2013b*), whereas *Albanensia* or a closely related form would occur in the latest Miocene of Shihuiba (Yunnan, China) (*Qiu, 2002*). Our phylogenetic analyses show that most extant Asian flying squirrel genera diverged during the interval from 15 to 10 Ma (*Figure 7* and *Figure 7—figure supplement 1*), thus coinciding with this time of high diversity and geographic range extension. On the contrary, 'flying' squirrel diversity in North America stayed at very low levels, with only the genera *Petauristodon* and *Sciurion* known from the middle and the beginning of the late Miocene (*Goodwin, 2008*) (*Figure 8*). Open habitats (mixed scrubland-grassland) housing a remarkable diversity of grazing mammals became widespread in central North America at that time, even though $C_4$-dominated grasslands did not spread until the late Miocene (ca. 7 Ma) (*Strömberg, 2011*). These unsuitable habitats would have hampered the radiation of flying squirrels there.

## Geographical range contraction and origins of extant flying squirrel genera

Throughout the later Miocene, warm-temperate forests continued reducing their extension and intermingled with drier and cooler biomes in Eurasia (*Pound et al., 2012*). In the Mediterranean regions, open habitats corresponding to woodlands and scrublands, already occurred in the southern Iberian Peninsula and Turkey since the middle Miocene (*Fauquette et al., 2007*; *Strömberg, 2011*), but would increase their extension at that time. Furthermore, the characteristic Mediterranean rainfall seasonality (summer drought) appeared during the Pliocene (*Suc, 1984*). In Central Europe, deciduous forests increasingly replaced warm-temperate mixed ones, which became restricted to coastal areas of the Mediterranean and the Parathetys (*Mosbrugger et al., 2005*). In Asia, major physiographical changes such as the uplift of the Himalaya-Tibetan Plateau affected atmospheric circulation and resulted in increased aridification of large areas of the continent (*Zhisheng et al., 2001*). Carbon stable isotope records of soil carbonates and ungulate tooth enamel indicate that $C_4$ grasses rise to dominance in ecosystems between 8 – 7 Ma in Pakistan (*Cerling et al., 1993*; *Strömberg, 2011*) and later in China (*Strömberg, 2011*). All these environmental changes had profound effects in flying squirrel diversity and biogeography. The group became increasingly rarer over much of its former range and several genera (*Forsythia*, *Albanensia*, *Miopetaurista*, *Blackia*) disappeared during the late Miocene and the Pliocene (*De Bruijn, 1999*) (*Figure 8*). In Asia, fossil occurrences during the late Miocene are mostly confined to the southeast (Yunnan province in China, Thailand) (*Lu et al., 2013*). In North America, only the species *Miopetaurista webbi* is known by very scarce remains from the latest Miocene and the Pliocene of Florida (*Goodwin, 2008*); *Robertson, 1976*), being the only record of this genus outside Eurasia (*Figure 8*). It is not surprising that this last record of a large-sized flying squirrel in North America comes from Florida, an area that is still characterized by a humid subtropical climate with abundant densely forested areas. The extant American flying squirrel genus *Glaucomys*, which today inhabits temperate

**Table 4.** Fossils used in the total evidence and node dating phylogenetic analyses.

*Miopetaurista neogrivensis* is considered as a calibration point for the divergence between *Petaurista* and all the other Pteromyina in node dating analysis and is considered as a tip date in the total evidence analysis. Root age in the total evidence analysis is calibrated considering the age of the oldest Aplodontiidae (sister group of Sciuridae), the oldest Gliridae (sister group of Aplodontiidae + Sciuridae), the oldest Myodonta, and the oldest-known rodent, the ischyromyid *Acritoparamys atavus*. The oldest sciurid that is reasonably complete (*Douglassciurus jeffersoni*) is used as tip date in the total evidence analysis and is also considered as a calibration point in the node dating analysis. See Appendix 1 for details on the calibration of the molecular clock using fossils.

| Fossil | Locality | Node on phylogeny | Max. age | Min. age | Taxonomic refs. | Age refs. |
|---|---|---|---|---|---|---|
| *Acritoparamys atavus* | Bear Creek, Montana, USA | maximum age for the root (oldest-known rodent) | 56.0 Ma | 55.8 Ma | *Janis et al., 2008; Korth, 1994* | *Janis et al., 2008* |
| *Erlianomys combinatus* | AS-1, Nuhetingboerhe, Erlian Basin, Inner Mongolia, China | maximum age for the root (oldest-known Myodonta) | 54.8 Ma | 53.9 Ma | *Li and Meng, 2010; Wu et al., 2012* | *Wu et al., 2012* |
| *Eogliravus wildi* | Mas de Gimel, Hérault, France | maximum age for the root (oldest-known Gliridae) | 50.7 Ma | 47.4 Ma | *Daams and Bruijn, 1995; Dawson, 2003; Hartenberger J-L and Hartenberger, 1971; Storch and Seiffert, 2007; Vianey-Liaud, 1985* | *Daams and Bruijn, 1995; Dawson, 2003* |
| *Spurimus selbyi* | Upper Teppe Formation, Wyoming, USA | minimum age for the root (oldest-known Aplodontiidae) | 46.3 Ma | 45.7 Ma | *Flynn and Jacobs, 2008; Korth, 1994* | *Flynn and Jacobs, 2008* |
| *Douglassciurus jeffersoni* | Renova Formation, Pipestone Springs, Montana, USA | *Douglassciurus*/other Sciuridae | 36.6 Ma | 35.8 Ma | *Emry and Korth, 1996; Emry and Thorington, 1982; Goodwin, 2008* | *Goodwin, 2008* |
| *Heteroxerus* sp. | Canales, Loranca Basin, Spain | Xerinae/Sciurinae | 26.4 Ma | 25.0 Ma | *Álvarez-Sierra et al., 1999* | *Álvarez-Sierra et al., 1999* |
| *Sciurus olsoni* | NB29A, Truckee Formation, Nevada, USA | *Sciurus/Tamiasciurus* | 13.6 Ma | 10.3 Ma | *Emry et al., 2005* | *Goodwin, 2008; Emry et al., 2005; Woodburne, 2004* |
| *Miopetaurista neogrivensis* | Abocador de Can Mata ACM/C5-D1, Vallès-Penedès Basin, Spain | *Petaurista*/other Pteromyina | | 11.6 Ma | this work | this work |

DOI: https://doi.org/10.7554/eLife.39270.022

deciduous forests and boreal coniferous forests, is recorded for the first time already in the Pleistocene (*Ruez, 2001*) although it would have diverged significantly earlier according to molecular results (*Figure 7* and *Figure 7—figure supplement 1*; *Arbogast et al., 2017*; *Mercer and Roth, 2003*).

The Pleistocene records of flying squirrels mostly correspond to extant genera and species (*Lu et al., 2013*) (*Figure 8*). This is again congruent with our results, which show that most extant species had already diverged during the Pliocene and some even at the latest Miocene (*Figure 7* and *Figure 7—figure supplement 1*). During the Pleistocene, *Glaucomys* is the only flying squirrel known from North America. In Asia, the genera *Aretes*, *Belomys*, *Petaurista*, *Pteromys* and *Trogopterus* have been recorded from several sites of south and eastern China (*Lu et al., 2013*). In addition, *Belomys* has also been reported from Thailand and *Pteromys* from Japan (*Lu et al., 2013*). The Pleistocene records of all these genera are located within their current geographical range and generally correspond to extant species. In the European Pleistocene, flying squirrels are represented by the genera *Neopetes*, *Petauria* and *Pliopetaurista* (*Figure 8*). Quite surprisingly, there are no fossil records of *Pteromys*, the only flying squirrel genus still extant in Europe. *Neopetes* comprises species formerly included in the extant genus *Hylopetes* (*Jackson and Thorington, 2012*). *Petauria* is a large-sized flying squirrel known from middle Pleistocene fissure fillings and cave deposits of Germany and Poland. The available cheek teeth show several striking morphological similarities with the

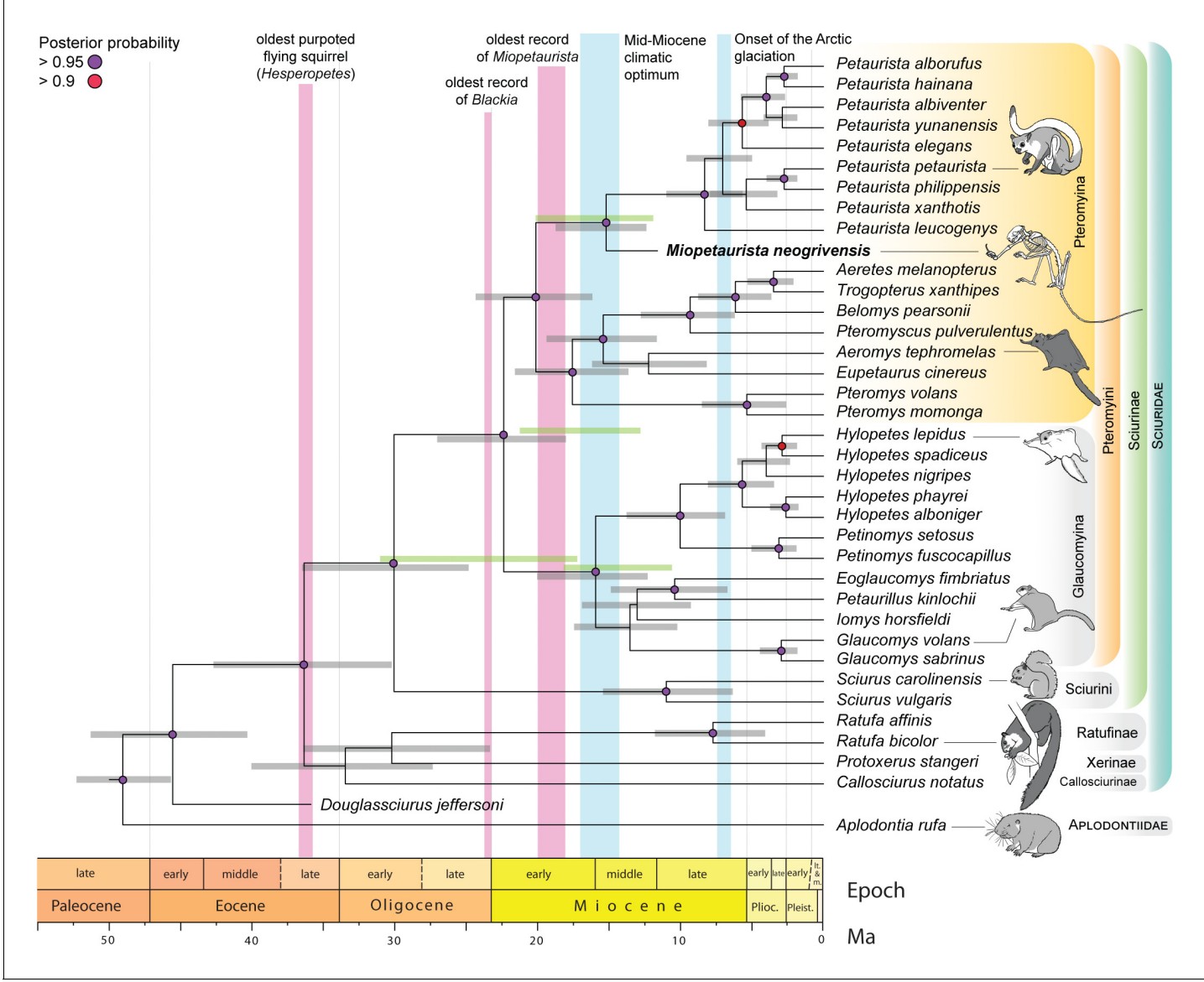

**Figure 7.** Flying squirrel phylogeny and node dating estimates based on a Bayesian total evidence analysis including *Miopetaurista neogrivensis*. The analysis is based on 38 taxa, 105 morphological characters and 3345 base pairs (see Materials and Methods and Appendix 1.1). Purple circles at the nodes indicate posterior probabilities higher than 0.95. Error bars (gray shading) at the nodes are 95% highest posterior density (HPD) intervals for divergence dates. For selected nodes, 95% HPD intervals derived from an independent node dating analysis using BEAST (*Figure 7—figure supplement 1*) are also shown as green bars. Note the position of *Miopetaurista neogrivensis* as sister taxon of extant *Petaurista*. The age of the oldest purported flying squirrels in the fossil record as well as that of the earliest representatives of the genera *Blackia* and *Miopetaurista* is indicated for comparison (see also *Figure 8*). Two major global climatic events are also indicated. The morphological character list is given in Appendix 2. Genbank accession numbers for all the sequences used in phylogenetic analyses are given in *Supplementary file 2* and morphological character matrix is given in *Supplementary file 3*.

DOI: https://doi.org/10.7554/eLife.39270.020

The following figure supplement is available for figure 7:

**Figure supplement 1.** Sciurinae phylogeny and node dating estimates using BEAST.

DOI: https://doi.org/10.7554/eLife.39270.021

extant *Petaurista*, including: smooth enamel with numerous lophules, particularly in the basin of the lower cheek teeth; prominent mesostylid in the lower molars; and presence of a well-developed postero-lingual re-entrant fold in the upper molars. We agree with the opinion of some authors that this genus is a junior subjective synonym of *Petaurista* (**Jackson and Thorington, 2012**;

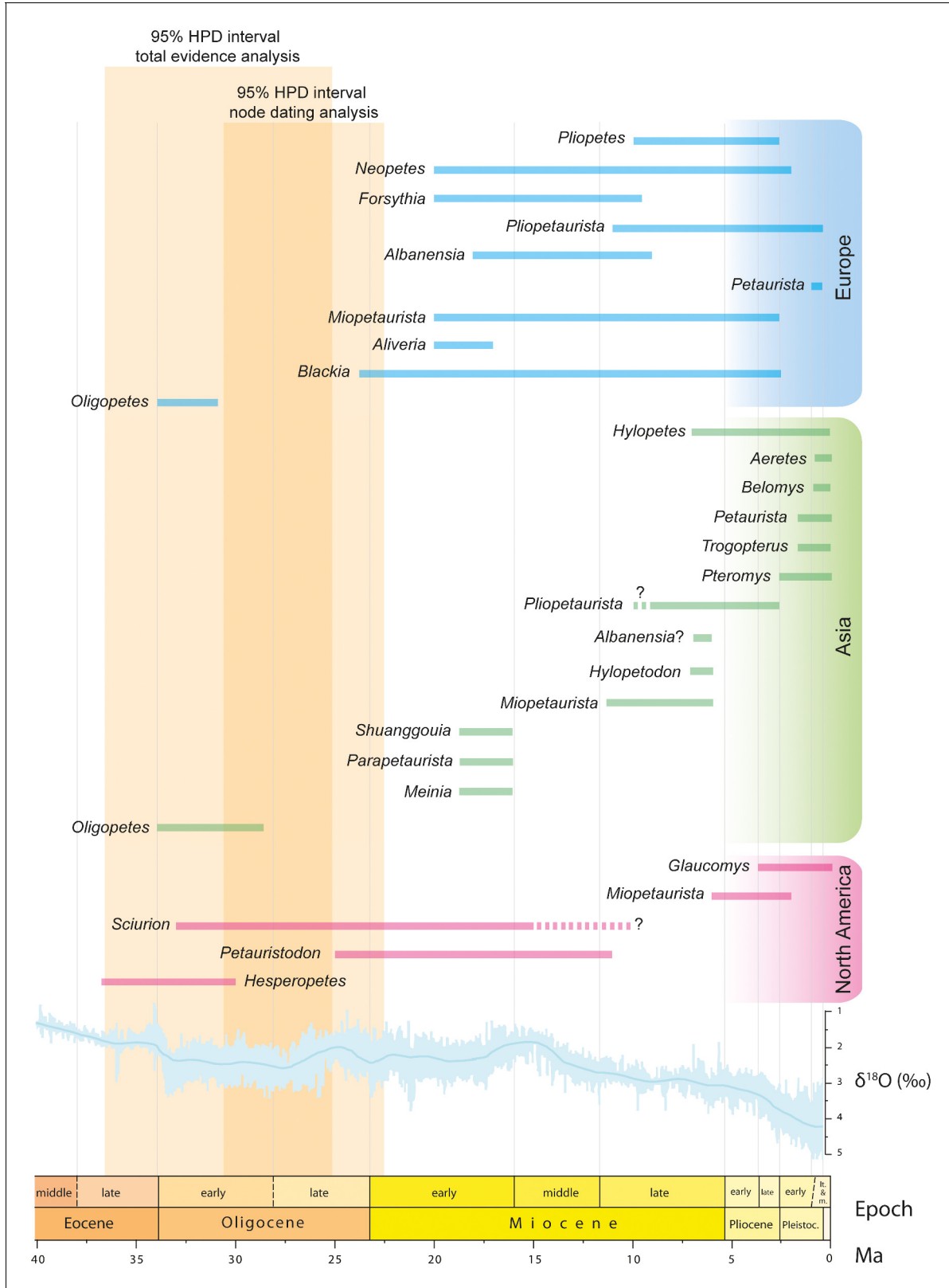

**Figure 8.** Fossil record of 'flying squirrels' and paleoclimatic data. Temporal ranges of purported flying squirrel genera in Europe, Asia and North America. The 95% highest posterior density (HPD) intervals for flying squirrel divergence as derived from total evidence and node dating analyses are indicated in orange shading (see *Figure 7* and *Figure 7—figure supplement 1* ). Darker shading indicates the time interval where both independently

*Figure 8 continued on next page*

*Figure 8 continued*

calculated estimates overlap, thus defining the most likely time interval for flying squirrel divergence. Global paleoclimatic data are taken from *Zachos et al., 2001*.

DOI: https://doi.org/10.7554/eLife.39270.023

*Thorington et al., 2005*). Therefore, the geographical range of *Petaurista* during the middle Pleistocene included Europe. Whether the genus originated in Europe or Asia cannot be resolved, since its closest relative, *Miopetaurista*, shows a similarly broad geographical range. It is worth noting that *Miopetaurista* is remarkably similar to *Petaurista*, to the point that their postcranial skeleton is virtually identical, even in specificities such as the more reduced lateral epicondylar ridge of the humerus or the wider patellar groove in the femur as compared to other (generally smaller) flying squirrels (for more detailed comparisons see Appendix 3.3). Cranial morphology evidences a close affinity between both genera, with details such as the development of the postorbital processes or the short and wide rostrum, being surprisingly similar. The only differences lie in cheek tooth morphology. *Miopetaurista* shows a relatively simple morphology, with faint enamel in the upper cheek teeth and no additional longitudinal lophules. In contrast, *Petaurista* presents relatively higher-crowned teeth and a much more complex morphology with additional longitudinal lophules (*Mein, 1970*; *Thorington et al., 2002*). In addition, the upper cheek teeth show a well-defined distolingual flexus, a feature that only occurs in the M3 in *Miopetaurista*. Dental morphological differences apart, it is worth noting that large-sized flying squirrels must be regarded as a very conservative group, having experienced little morphological changes since the late middle Miocene.

Concerning the genus *Petaurista*, our analyses recognize the three main species groups (*Figure 7* and *Figure 7—figure supplement 1*) that had already been recognized by previous molecular analyses (*Li et al., 2013*). The first group solely includes *Petaurista leucogenys*, a species endemic to Japan. It is found to have diverged during the late Miocene (ca. 11 – 6 Ma), significantly earlier than the remaining species. The second clade includes *Petaurista petaurista* and *Petaurista philippensis*, the most widely distributed species in the genus, together with *Petaurista xanthotis*, endemic to south-central China and Tibet. Finally, the third clade comprises species mostly occurring in southern and central China (*Petaurista albiventer*, *Petaurista alborufus*, *Petaurista yunanensis*) and Indonesia (*Petaurista elegans*), with *Petaurista hainana*, which is endemic to Hainan Island (southeastern China). These two *Petaurista* subclades diverged during the late Miocene and estimates for the divergence of most species range from the late Pliocene to the early Pleistocene, that is coinciding with global cooling after 3 Ma and the start of northern hemisphere glaciation (*Zachos et al., 2001*). Other flying squirrel genera, such as *Hylopetes*, *Petinomys* and *Glaucomys* show a similar pattern, with estimates for the divergence of many extant species ranging from the late Pliocene to the early Pleistocene.

In the present, flying squirrels are widely distributed across the northern hemisphere, but only one genus lives in North America (*Glaucomys*) and a single species (*Pteromys volans*) occurs in Europe. Very few species are endemic to central and northern Asia. In contrast, flying squirrels are diverse in the tropical and subtropical forests of the Indo-Malayan region, which apparently acted both as refugia and diversification center (also for tree squirrels) since the late Miocene (*Lu et al., 2013*; *Mercer and Roth, 2003*).

## Conclusions

*Miopetaurista neogrivensis* is the oldest unquestionable flying squirrel and dates back to the middle/late Miocene boundary (11.6 Ma). Its diagnostic wrist anatomy indicates that the two subtribes of flying squirrels had already diverged at that time. Moreover, this new fossil allows for a recalibration of flying squirrel time of origin and diversification, generally providing somewhat older estimates than previous molecular analyses. These differ according to the phylogenetic method used, total evidence analysis estimates an interval of 36.6 – 24.9 Ma while node dating results in a younger estimate of 30.6 – 17.4 Ma. Therefore, we cannot rule out that at least some of the oldest (ca. 36 Ma) fossils tentatively identified as flying squirrels may indeed belong to this group. However, the estimates of both independent phylogenetic approaches overlap for the late Oligocene (31 – 25 Ma), which should be considered the most likely interval for flying squirrel divergence. The two flying

squirrel subtribes are found to have diverged during the early Miocene (22 – 18 Ma) while most extant genera would do so during the Miocene, although they are not recorded until the Pleistocene. *Miopetaurista neogrivensis* is estimated to have diverged from *Petaurista* spp., its sister taxon, between 18.8 – 12.4 Ma, the oldest boundary overlapping with the earliest record of the genus *Miopetaurista* (18 – 17 Ma). Perhaps not surprisingly, the skeletons of both genera show little differences. Sciurids are often regarded as a morphologically conservative group and flying squirrels are no exception having experienced few morphological changes for almost 12 million years.

### Important information on data availability

3D surface models of the described material of *M. neogrivensis* (reconstructed skeleton, reconstructed skull, carpal bones) are available at MorphoBank https://morphobank.org/index.php/Projects/ProjectOverview/project_id/3108. All other data files are provided as supplementary data.

## Materials and methods

### Provenance and chronology

The partial skeleton of *Miopetaurista neogrivensis* (IPS56468) and all other described material (*Figure 2—figure supplement 1-2*) is housed in the Institut Català de Paleontologia Miquel Crusafont (ICP) in Sabadell (Barcelona, Catalonia, Spain). IPS56468 was found in 2008 during paleontological surveillance of excavation works at the Can Mata landfill (Abocador de Can Mata [ACM], els Hostalets de Pierola, Catalonia, Spain). It was unearthed from a rich fossiliferous horizon ACM/C5-D1 (ACM Cell 5, sector D, locality 1) in two different blocks with some elements in anatomical connection (*Figure 2*; *Figure 7—figure supplement 1*). The excavation of ACM/C5-D1 provided additional material of *M. neogrivensis* (*Table 2*). A partial, dorsoventrally crushed skull (IPS88677) was also recovered from stratigraphically close locality ACM/C8-Af (ACM Cell 8, sector A, locality f) (*Alba et al., 2017*).

The ACM series is located in the Vallès-Penedès Basin, an elongated half-graben filled mostly by continental deposits during the Miocene, which are rich in vertebrate fossils (*Figure 9*). The ACM composite series ranges from ca. 12.6 to 11.5 Ma, the age of the paleontological localities being well constrained thanks to high-resolution litho-, bio- and magnetostratigraphical data (*Alba et al., 2017*). ACM/C5-D1 is correlated to chron C5r.2n (11.657 – 11.592 Ma) (*Alba et al., 2017*), and its interpolated age is 11.64 Ma (*Alba et al., 2017*).

### Anatomical terminology

Dental terminology, abreviations and measurement methods for sciurid cheek teeth follow *Casanovas-Vilar et al., 2015* and references therein.

### Three-dimensional data acquisition and reconstruction

A virtual three-dimensional (3D) cranial reconstruction of *M. neogrivensis* was performed based on both IPS56468h and IPS88677 (*Figure 6—figure supplement 1-2*, *Video 3*). The specimens were analyzed separately by microfocus X-ray computed tomography (µCT) at the Multidisciplinary Laboratory of the 'Abdus Salam' International Centre of Theoretical Physics (Trieste, Italy), using a system specifically designed for the study of archaeological and paleontological materials (*Table 5*). Raw data from each scanning were imported (as stack of TIFF 8-bit files) to Avizo 7.0 and Rhinoceros 5.0 for segmentation, repositioning, mirroring and visualization. Each cranial bone or bone fragment was segmented virtually removing the surrounding matrix using semiautomatic thresholding tools and obtaining individual 3D digital models. Up to 64 3D models were generated for both specimens prior to repositioning and mirroring them to assemble an almost complete skull. Reconstruction primarily relied upon IPS56468h, but used 3D bone models from IPS88677 for elements that were particularly damaged or missing from the former (*Table 3*). Due to poor preservation of some fragments, it was necessary to import some of the 3D models to Rhinoceros 5.0 to repair the meshes, split the model to keep only the well-preserved regions, and use, if available, the mirrored region from the other side of the same skull, or alternatively take it from IPS88677 (mirrored if necessary). The 3D models were repositioned using Avizo 7.0 based on bilateral symmetry and fracture congruence. An almost complete 3D virtual skull model was finally assembled using a total of 41 3D

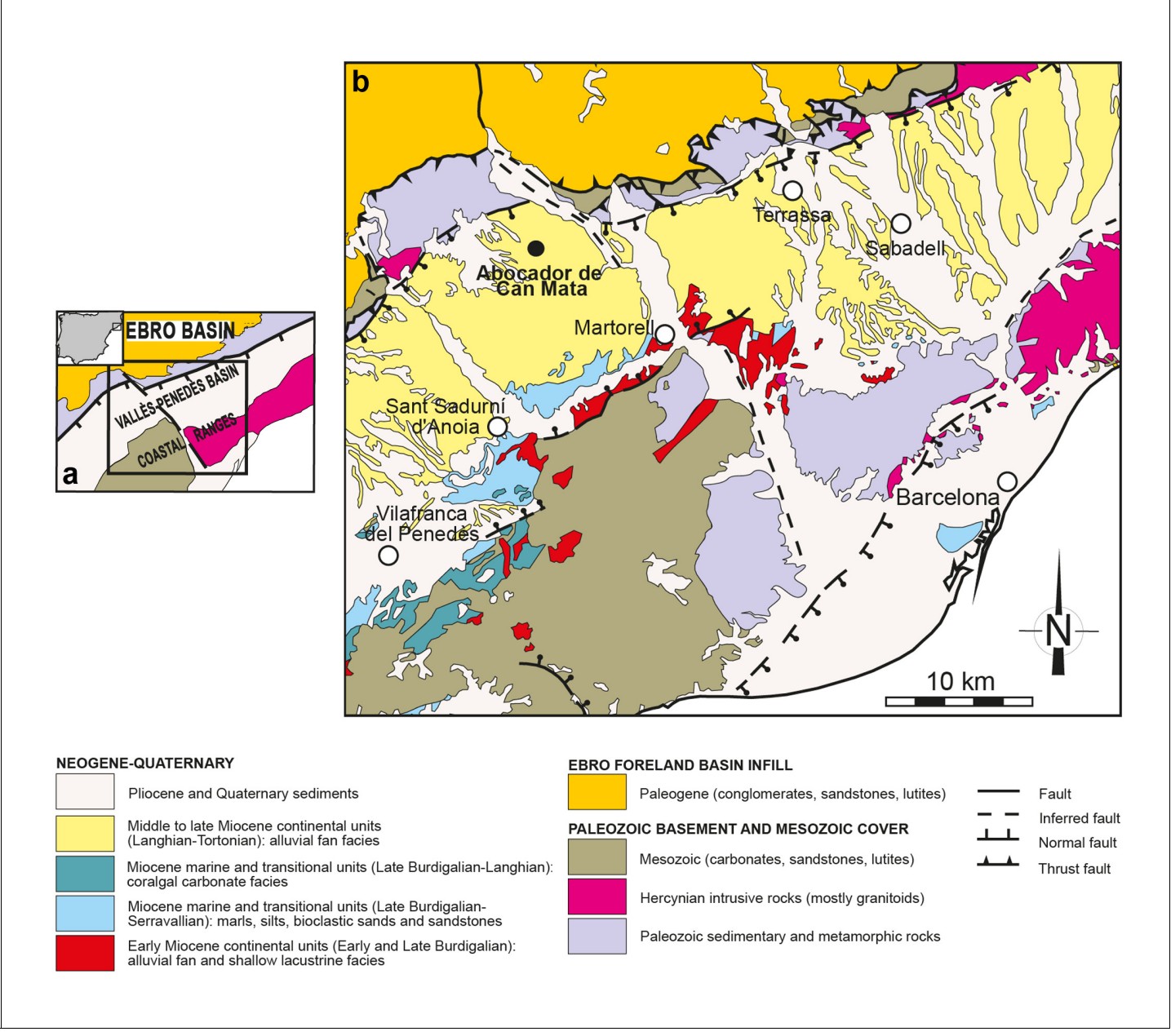

**Figure 9.** Geological map of the Vallès-Penedès Basin and situation of the fossil site. (a) Schematic geological map of the Vallès-Penedès Basin (Catalonia, Spain) showing the area enlarged in *Figure 9b*. The inset shows its location within the Iberian Peninsula. (b) Situation of the Abocador de Can Mata series, located in distal alluvial fan facies of middle to late Miocene age. See *Alba et al., 2017* for further details on the stratigraphy and chronology of the Abocador de Can Mata series and main sites. Map modified from *Casanovas-Vilar et al., 2016*.
DOI: https://doi.org/10.7554/eLife.39270.024

models of cranial bones (*Figure 6*, *Table 3*). Besides cranial restoration, the two mandibles belonging to IPS56468 were scanned using a 3D desktop laser scanner (NextEngine) at high definition and with a dimensional accuracy of 0.13 mm (Macro Mode). Mandibles were exported to Rhinoceros 5.0 to repair meshes and were repositioned and joined to the resulting skull model using Avizo 7.0.

In addition, the posterior region of the skull of the extant flying squirrel *Petaurista petaurista* was scanned to compare its inner ear morphology to that of *M. neogrivensis* (*Video 4*). The specimen is kept at the Naturalis Biodiversity Center (NBC, Leiden, the Netherlands) and was scanned at this

**Table 5.** μCT scan parameters used for *Miopetaurista neogrivensis* and additional comparative material.
IPS, acronym for the collections of the Institut Català de Paleontologia Miquel Crusafont. All other specimens are kept in the collections of the Naturalis Biodiversity Center (Leiden, the Netherlands). When a filter was used, it was aluminum. For additional details see Materials and Methods.

| Catalogue no. | Taxon | Element | Voltage (kV) | Current (μA) | Filter | Projections | Voxel size (μm) |
|---|---|---|---|---|---|---|---|
| IPS56468h | *Miopetaurista neogrivensis* | skull | 130 | 72 | 1 mm | 1880 | 39.71 |
| IPS88677 | *Miopetaurista neogrivensis* | skull | 130 | 72 | 1 mm | 1880 | 39.71 |
| ZMA13418 | *Petaurista petaurista* | skull | 100 | 100 | 0.5 mm | 1290 | 13.17 |
| IPS56468ah | *Miopetaurista neogrivensis* | carpals | 80 | 124 | not used | 651 | 10.67 |
| 056.054.b | *Hylopetes sagitta* | carpals | 100 | 100 | not used | 558 | 11.32 |
| RMNH.MAM.130 | *Petaurista petaurista* | carpals | 79 | 125 | not used | 482 | 13.17 |
| 27291 | *Sciurus vulgaris* | carpals | 59 | 167 | not used | 550 | 13.04 |

DOI: https://doi.org/10.7554/eLife.39270.025

institution with a μCT scanner, using a Skyscan system model 1172 (Bruker company, Belgium) (*Table 5*).

The 3D surface of the postcranial skeleton of *M. neogrivensis* was digitized by means of photogrammetry using a variable number of sets of two-dimensional images. Within each set, individual images were captured with a minimum overlap of 66% using a digital single-lens reflex camera with a 50 mm f/2 lens. Aperture and focus were constant to ensure consistency between sets of photographs. Each set was then aligned and scaled to create a 3D model using the photogrammetric software Agisoft PhotoScan Professional. Individual surface models were exported to the 3D software Pixologic ZBrush version 4R7, which was used to merge the different meshes and improve the quality of the final merged 3D surface. Missing areas were modeled according to mirrored bones when available. Missing bone elements and broken areas in the available material were reconstructed using *P. petaurista* as reference. The morphology of the axial skeleton follows that of *P. petaurista* but has been modified according to the proportions of the recovered vertebrae.

Finally, carpal bones of extant *Sciurus vulgaris*, *P. petaurista* and *Hylopetes sagitta*, as well as those from *M. neogrivensis* were digitized for visualization and comparison. All extant specimens are stored in the collections of the NBC and were scanned there using a μCT scanner Skyscan system model 1172 (*Table 5*). Raw CT data from each scanning were imported directly (as stack of TIFF 16-bit files) to Avizo 7.0.

## Body mass estimation

Body mass (BM, in g) was estimated using an allometric regression of BM vs. skull length (SL, in mm), which is the estimator most tightly correlated with BM in extant rodents (*Bertrand et al., 2016*). SL was measured in *M. neogrivensis* from the 3D reconstruction based on crania IPS56468h and IPS88677 simultaneously (69.80 mm). The allometric regression was performed using log-transformed (ln) mean species data, as it is customary when predicting BM (*Ruff, 2003*). Sexes were treated separately (when possible) to avoid potentially confounding effects of body size dimorphism. The regression was computed for sciurids only to avoid biases due to allometric grade shifts among rodent families. SL sex-species means were calculated from published individual values (*Bertrand et al., 2016*) or measured by one of the authors (I.C.V.). BM average data were computed from individual published values (*Bertrand et al., 2016*) or taken from the literature (*Hayssen, 2008*; *Thorington and Heaney, 1981*; *Zahler, 2001*). The ordinary least-squares regression method, selected as the most suitable for prediction (*Ruff, 2003*; *Smith, 1994*), yielded the following equation with SPSS v. 17: ln BM = 4.369 ln SL – 11.384 (N = 33, p<0.001, r = 0.972, SEE = 0.256). The logarithmic detransformation bias was corrected using the quasimaximum likelihood estimator (*Ruff, 2003*; *Smith, 1993*): QMLE = 1.033. 50% confidence intervals (CI) for the prediction were computed.

## Phylogenetic analyses and divergence dating

To infer the phylogenetic placement of *M. neogrivensis* and assess its impact on the dating estimates of the phylogeny of flying squirrels, we performed a new phylogenetic analysis of tribe Pteromyini by means of two alternative approaches: total evidence (*Ronquist et al., 2012*) and node dating (*Ronquist et al., 2016*). These approaches relied on the combination of morphological and molecular datasets, or on molecular datasets alone, respectively.

For the molecular dataset, we retrieved from GenBank the combination of the four genes more frequently used in previous phylogenetic studies of the Sciuridae (12S, 16S, cytochrome b, and irbp) (*Mercer and Roth, 2003*), which were downloaded for all Sciurinae species available in GenBank (*Supplementary file 2*). This dataset comprises 58 of the 89 extant Sciurinae species (*Koprowski et al., 2016*) and all genera but *Biswamoyopterus*. The Pteromyini are represented by 29 species, comprising 65% of extant diversity. We also obtained the sequence for *Aplodontia rufa*, sole extant member of the Aplodontiidae (the sister group of the Sciuridae; *Fabre et al., 2012*; *Huchon et al., 2002*), which was used as outgroup in the total evidence analysis. Each gene was aligned using two procedures: ribosomal coding genes were aligned by means of MAFFT v6 (*Katoh et al., 2002*) and protein coding genes were aligned using the translation alignment algorithm implemented in TranslatorX (using MAFFT to align proteins) (*Abascal et al., 2010*). Poorly-aligned regions in the ribosomal coding genes were eliminated with Gblocks (*Castresana, 2000*) under low stringency options (*Talavera and Castresana, 2007*). The morphological dataset consisted of 105 characters comprising the dentition, skull and postcranial skeleton of 36 extant taxa plus two fossil squirrels: *M. neogrivensis* and *Douglassciurus jeffersoni* (Appendix 2; *Supplementary file 3*). *Douglassciurus* is chosen because it is the earliest squirrel represented by abundant postcranial material and is currently recognized as an outgroup to all other sciurids (Appendix 1.1). The character list is mostly based in *Thorington et al., 2002* with numerous additions of diagnostic pteromyin characters from the limb bones after *Thorington et al., 2005*. Finally, a set of new characters was also added to resolve the relationships of *A. rufa* and *D. jeffersoni* with other taxa (Appendix 2, *Supplementary file 3*).

For the total evidence dating we used MrBayes v. 3.2.6 (*Ronquist et al., 2012*). The analysis included 36 species for which we had both molecular and morphological data plus two fossil species with only morphological data. The molecular data consisted in a total of 3345 base pairs (bps) distributed in each gene as follows: 12S (505 bps), 16S (521 bps), cytb (1140 pbs), irbp (1179 pbs). The protein coding genes were split in codon positions and the most appropriate partitioning scheme and the model of molecular evolution for each partition were estimated by means of the software Partitionfinder (*Lanfear et al., 2012*) (*Table 6*). The five partitions derived from Partitionfinder were then concatenated with the previously described morphological dataset. We modeled node ages

**Table 6.** Model specifications for phylogenetic analyses.
Molecular model specifications for total evidence and node dating phylogenetic analyses.

| Analysis software | Genes | Model | Gamma distribution | Invariant |
|---|---|---|---|---|
| BEAST | cytb codon 1, 16S, 12S | GTR | yes | yes |
| BEAST | cytb codon 2, irbp codon 2 | HKY | no | yes |
| BEAST | cytb codon 3 | GTR | yes | yes |
| BEAST | irbp codon 1 | HKY | no | no |
| BEAST | irbp codon 3 | HKY | yes | no |
| MrBayes | cytb codon 1, 16S, 12S | GTR | yes | yes |
| MrBayes | cytb codon 2, irbp codon 2 | HKY | yes | yes |
| MrBayes | cytb codon 3 | GTR | yes | yes |
| MrBayes | irbp codon 1 | HKY | yes | no |
| MrBayes | irbp codon 3 | GTR | yes | no |

DOI: https://doi.org/10.7554/eLife.39270.026

and tree topology using a fossilized birth and death process (*Gavryushkina et al., 2017*; *Heath et al., 2014*; *Stadler, 2010*; *Zhang et al., 2016*) with broad priors on speciation (exp[10]), extinction (beta[1,1]), and fossilization (beta[1,1]) (*Pyron, 2017*; *Zhang et al., 2016*).The analysis also relied on a relaxed-clock with the independent gamma-rates (IGR) model (*Ronquist et al., 2012*), with a broad prior for the variance increase parameter (exp[10]; *Pyron, 2017*). Following Pyron (*Pyron, 2017*) we used 1/mean of the root age and exp(1/mean of the root age) to generate a wide clock rate prior compatible with both morphological and molecular clocks. For the root age we used the interval 52.7–45.7 Ma as a uniform prior. The lower bound of this prior was informed by the most recent age estimate (MRE) of the oldest Aplodontiidae (*Table 4*; see also Appendix 1.1) and we calculated the upper bound using the algorithm proposed by *Hedman, 2010*, which calculates the probability distribution of the age of a clade, given the ages of the oldest fossil representatives of the outgroups of that clade. We used the upper limit of the 95% confidence interval for the age of the root as a plausible upper bound for the root. This was informed by the MRE for the minimum age of the oldest Gliridae (47.4 Ma, outgroup to Aplodontiidae+ Sciuridae (*Blanga-Kanfi et al., 2009*; *Fabre et al., 2012*; *Huchon et al., 2002*) and the MRE for the minimum age of *Erlianomys combinatus*, the oldest Myodonta (53.9 Ma, outgroup to Gliridae +Aplodontiidae + Sciuridae; *Blanga-Kanfi et al., 2009*; *Fabre et al., 2012*; *Huchon et al., 2002*); *Table 4*, Appendix 1.1). The maximum age oldest rodent was used as the maximum possible age of the root (56 Ma; *Table 4*, Appendix 1.1).

Partitions and nucleotide substitution models for the molecular data were estimated by means of Partitionfinder v. 2.1.1 (*Lanfear et al., 2012*) with linked branch lengths, a Bayesian Information Criterion (BIC) model of selection and a greedy search algorithm (*Table 6*). For the morphological data (only variable characters included), we used a k-state Markov (Mkv) model (*Lewis, 2001*) with a rate variation modeled by means of a discrete gamma model. We distinguished between ordered and unordered characters.

We also used an alternative Bayesian program, BEAST v. 1.8.4 (*Drummond et al., 2012*), to provide an independent estimate of the time of the Pteromyini/Sciurini divergence and the onset of Pteromyini crown diversification by means of node dating. This analysis included all species of Sciurinae available in GenBank (58 taxa) and consisted in 3225 pbs distributed in each gene as follows: 12S (383 bps), 16S (523 bps), cytb (1140 pbs), irbp (1179 pbs). Protein coding genes were subsequently split in codon positions and we used Partitionfinder to determine the best set of molecular partitions and models of molecular evolution (*Table 6*). We estimated the tree in time units using an uncorrelated lognormal clock applied to each of the five partitions derived from Partitionfinder and two different calibration points (one located in each tribe; *Table 4*). *Miopetaurista neogrivensis* from ACM/C5-D1 (11.6 Ma) provides a minimum age for the divergence between *Petaurista* and the remaining Pteromyina, while the oldest record of the genus *Sciurus* (dating back to 13.6–10.3 Ma) gives minimum age for the *Sciurus/Tamiasciurus* split (*Table 4*, Appendix 1.2). Soft maxima for both calibrations were calculated as described in *Hedman, 2010*. The soft maximum for the calibration point in the Pteromyina was informed by the MRE of the oldest *Heteroxerus*, (25.0 Ma), the earliest Xerinae (sister group to Sciurinae), and by the oldest estimate of *D. jeffersoni* (35.8 Ma), basal to crown Sciuridae (*Table 4*, Appendix 1.2). For the calibration point in the Sciurini, the soft maximum was informed by the age of *M. neogrivensis* and the ages of oldest *Heteroxerus* and *D. jeffersoni* stated above. Following the 'consistent approach' (*Hedman, 2010*) the minimum age of the oldest Aplodontiidae (45.7 Ma) is taken as the maximum possible age of both calibration points.

Both MrBayes and BEAST analyses relied on four independent runs of 25,000,000 generations, each sampled every 4000 generations. Each of the runs in MrBayes relied on four Markov chain Monte Carlo (MCMC) chains.

## Acknowledgements

This research has been funded by the Spanish Agencia Estatal de Investigación and the European Regional Development Fund of the European Union (CGL2016-76431-P, CGL2017-82654-P, RYC-2013 – 12470 to IC-V. FJCI-2014 – 20380 fellowship to JG-P.), by the Generalitat de Catalunya (CERCA Programme), and by the Handel T Martin Endowment to O S. We further acknowledge the support of the Servei d'Arqueologia i Paleontologia of the Generalitat de Catalunya. Fieldwork was defrayed by CESPA Gestión de Residuos, S A U. We thank M Valls (ICP) for the excellent preparation

of the specimens, which was partly funded by the Department of Cultura of the Generalitat de Catalunya (2014/100609). The authors thank L W van den Hoek Ostende and W Renema for allowing access to the Naturalis Biodiversity Center (Leiden, the Netherlands) µCT facility, S van der Mije and P Kaminga for their assistance during the study of the squirrel collections kept at this institution. S Moyà-Solà (ICP), L J Flynn and D Pilbeam (Harvard University) provided valuable discussions on the fossil material and comments on the manuscript. We thank J Madurell-Malapeira (ICP) for his assessment in the preparation of some of the figures and J X Samuels (East Tennessee State University) for providing valuable data regarding the North American 'flying squirrel' record. Finally, we thank the editors and reviewers of this work for their valuable and constructive comments.

## Additional information

### Funding

| Funder | Grant reference number | Author |
|---|---|---|
| Agencia Estatatal de Investigación and European Regional Development Fund | CGL2017-82654-P | Isaac Casanovas-Vilar<br>Josep Fortuny |
| Generalitat de Catalunya | CERCA Programme | Isaac Casanovas-Vilar<br>Joan Garcia-Porta<br>Josep Fortuny<br>Marina Querejeta<br>Sergio Llácer<br>Josep M Robles<br>David M Alba |
| Handel T. Martin Endowment | | Óscar Sanisidro |
| Agencia Estatal de Investigación and European Regional Development Fund | CGL2016-76431-P | Isaac Casanovas-Vilar<br>Josep M Robles<br>David M Alba |
| Agencia Estatal de Investigación and European Regional Development Fund | RYC-2013-12470 | Isaac Casanovas-Vilar |
| Agencia Estatal de Investigación and European Regional Development Fund | FJCI-2014-20380 | Joan Garcia-Porta |

The funders had no role in study design, data collection and interpretation, or the decision to submit the work for publication.

### Author contributions

Isaac Casanovas-Vilar, Conceptualization, Resources, Data curation, Formal analysis, Supervision, Funding acquisition, Investigation, Visualization, Writing—original draft, Project administration, Writing—review and editing; Joan Garcia-Porta, Conceptualization,Resources, Data curation, Software, Formal analysis, Funding acquisition,Validation, Investigation, Visualization, Methodology, Writing—original draft; Josep Fortuny, Resources, Data curation, Formal analysis, Supervision, Investigation, Visualization, Methodology, Writing—original draft; Óscar Sanisidro, Formal analysis, Investigation, Writing—original draft; Jérôme Prieto, Resources, Data curation, Formal analysis, Validation; Marina Querejeta, Data curation, Formal analysis; Sergio Llácer, Josep M Robles, Data curation, Software, Formal analysis; Federico Bernardini, Formal analysis, Supervision, Funding acquisition, Investigation, Methodology, Writing—original draft, Writing—review and editing; David M Alba, Resources, Data curation,Funding acquisition, Visualization, Methodology, Writing—original draft

### Author ORCIDs

Isaac Casanovas-Vilar (iD) http://orcid.org/0000-0001-7092-9622
Joan Garcia-Porta (iD) https://orcid.org/0000-0003-4032-9495
Josep Fortuny (iD) https://orcid.org/0000-0003-4282-1619
Óscar Sanisidro (iD) http://orcid.org/0000-0002-8238-6394

Jérôme Prieto [iD] http://orcid.org/0000-0002-7485-9112
Sergio Llácer [iD] https://orcid.org/0000-0003-0192-7943
Josep M Robles [iD] http://orcid.org/0000-0002-5410-3529
Federico Bernardini [iD] http://orcid.org/0000-0002-3282-8799
David M Alba [iD] http://orcid.org/0000-0002-8886-5580

**Decision letter and Author response**
Decision letter https://doi.org/10.7554/eLife.39270.041
Author response https://doi.org/10.7554/eLife.39270.042

## Additional files

**Supplementary files**

• Supplementary file 1. Reconstruction of the skeleton of *Miopeaturista neogrivensis.*
DOI: https://doi.org/10.7554/eLife.39270.027

• Supplementary file 2. GenBank accession numbers for all the sequences used in phylogenetic analyses. Accession numbers for all the species considered in total evince analysis and/or node dating analysis. We retrieved the four genes more frequently used in previous phylogenetic studies of the Sciuridae: 12S, 16S, cytochrome b, and irbp (*Mercer and Roth, 2003*).
DOI: https://doi.org/10.7554/eLife.39270.028

• Supplementary file 3. Morphological character matrix for total evidence phylogenetic analysis. See Appendix 2 for character description and coding.
DOI: https://doi.org/10.7554/eLife.39270.029

• Supplementary file 4. Cheek tooth measurements of *Miopetaurista neogrivensis.* Cheek tooth measurements of *Miopetaurista neogrivensis* from localities ACM/C5-D1, ACM/C8-Af, ACM/C6-A5, as well as sector ACM/C8-B (material is listed in *Tables 1–2*). Measurement method follows *Casanovas-Vilar et al., 2015*: L, mesiodistal length; W, buccolingual width; w1, buccolingual width measured at the mesial margin of the tooth (only for m1 and m2); w2, buccolingual width measured at the distal margin of the tooth (only for m1 and m2). Estimated measurements (due to minor damage to the cheek teeth) are within parentheses. Abbreviations for dental elements follow *Smith and Dodson, 2003*. IPS, acronym for the collections of the Institut Català de Paleontologia Miquel Crusafont. All measurements are in mm.
DOI: https://doi.org/10.7554/eLife.39270.030

• Supplementary file 5. Cranial measurements of *Miopetaurista neogrivensis* and extant flying squirrels. Cranial measurements for *Miopetaurista neogrivensis* and some extant flying squirrels. The specimens of the extant species are housed in the Naturalis Biodiversity Center (Leiden, the Netherlands), except for *Glaucomys sabrinus*, which comes from the collections of the Institut Català de Paleontologia Miquel Crusafont. Cranial measurements for the virtually reconstructed skull of *M. neogrivensis* are also included. Estimated measurements are in parentheses. Acronyms and definitions of the measurements follow *Bertrand et al., 2016* and *Nicolas et al., 2008*: BNAS, maximum breadth of the nasals; BRCA, maximum breadth of the braincase; BULL, maximum length of the auditory bulla; CTL, length of the upper cheek tooth row; DIA, length of the diastema (from the alveolus of the incisor to the alveolus of the P3); HEBA, henselion–basion length; HEPA, henselion–palation length; IF, length of the incisive foramen; INTE, smallest interorbital breadth; LNAS, maximum length of the nasals; PALA, smallest palatal breadth; RB, maximum rostrum breadth; RH, mediosagittal projection of rostrum height at the anterior border of the M1; SL, maximum length of the skull. The virtual reconstruction of the cranium of *M. neogrivensis* is shown in *Figure 6*, for the original crania see *Figure 6—figure supplements 1–2*. All measurements are in mm.
DOI: https://doi.org/10.7554/eLife.39270.031

• Supplementary file 6. Mandibular measurements of *Miopetaurista neogrivensis* and extant flying squirrels. Mandibular material of *Miopetaurista neogrivensis* is listed in *Tables 1–2*. The specimens of the extant species are housed in the Naturalis Biodiversity Center (Leiden, the Netherlands), except for *Glaucomys sabrinus* which comes from the collections of the Institut Català de Paleontologia Miquel Crusafont. Estimated measurements are in parentheses. Acronyms and definitions of the measurements are as follows: DIA1, length of the diastema measured from the alveolus of the incisor

to the anterior alveolus of the p4; DIA2, length of the diastema from the anterior alveolus of the p4 to the projection of the incisor; DDIA, depth of the mandibular corpus at the lowermost point of the diastema; DM1, depth of the mandibular corpus below the anterior border of the m1; DM3, depth of the mandibular corpus below the anterior border of the m3; LAJ, length from the most ventral point of the angular apophysis to the anterior margin of the articular condyle; LCJ, length from the tip of the coronoid apophysis to the anterior margin of the articular condyle; LTL, length of the lower cheek tooth row; ML, maximum length of the mandible; RH, maximum height of the mandibular ramus. The mandibular remains of *M. neogrivensis* are shown in *Figure 3*. All measurements are in mm.

DOI: https://doi.org/10.7554/eLife.39270.032

• Supplementary file 7. Postcranial measurements of *Miopetaurista neogrivensis* and extant squirrels. Postcranial measurements for *Miopetaurista neogrivensis* (for material list see *Table 1*) and some extant squirrels. The specimens of the extant species are housed in the Naturalis Biodiversity Center (Leiden, the Netherlands), except for *Glaucomys sabrinus* and *Sciurus carolinensis* which come from the collections of the Institut Català de Paleontologia Miquel Crusafont. Only right elements were measured. Acronyms and definitions of the measurements are based in *Samuels and Van Valkenburgh, 2008*: BI, brachial index (RL/HL), indicating the proportions of the distal and proximal elements of the forelimb; CI, crural index (TL/FL), indicating the proportions of the distal and proximal elements of the forelimb; FEB, epicondylar mediolateral breadth of the distal femur; FGT, mediolateral breadth of the greater trochanter of the femur; FL, femoral length from the tip of the femoral head to the tip of the medial epicondyle; FMD, greater diameter of the femur at midshaft; FRI, femoral robusticidty index (FMD/FL); HEB, epicondylar mediolateral breadth of the distal humerus; HL, humeral length from the tip of the head to the tip of the most distal point of the trochlea; HMD greater diameter of the humerus at midshaft; HRI, humeral robusticity index (HMD/HL); IM, intermembral index [(HL +RL)/(FL +TL)], indicating the length of the forelimb relative to the hindlimb; TL, tibial length from the medial condyle to the tip of the medial malleolus; TMD, greater diameter of the tibia at midshaft; TRI, tibial robusticity index (TMD/TL); RL, radius lenght. All measurements are in mm. Specimens shown in *Figure 5* are: *Miopetaurista neogrivensis* IPS56468; *Petaurista petaurista* RMNH.MAM.130; *Hylopetes sagitta* 056.054.b; *Callosciurus prevostii* 056.099 .j; *Urocitellus paryii* 065.238a; *Xerus erythropus* 26483.

DOI: https://doi.org/10.7554/eLife.39270.033

• Transparent reporting form

DOI: https://doi.org/10.7554/eLife.39270.034

## Data availability

3D surface models of the described material of M. neogrivensis (reconstructed skeleton, reconstructed skull, carpal bones) are available at MorphoBank with identifier http://morphobank.org/permalink/?P3108. GenBank accession numbers for all the extant taxa considered in the phylogenetic analyses are provided in Supplementary File 2. All other data files are included in the supplementary information.

The following dataset was generated:

| Author(s) | Year | Dataset title | Dataset URL | Database, license, and accessibility information |
|---|---|---|---|---|
| Isaac Casanovas-Vilar, Joan Garcia-Porta, Josep Fortuny, Óscar Sanisidro, Jérôme Prieto, Marina Querejeta, Sergio Llácer, Josep M Robles, Federico Bernardini, David M Alba | 2018 | 3D surface models of the described material of Miopetaurista neogrivensis (reconstructed skeleton, reconstructed skull, carpal bones) | https://morphobank.org/index.php/Projects/ProjectOverview/project_id/3108 | Publicly available at Morphobank |

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

## Appendix 1

DOI: https://doi.org/10.7554/eLife.39270.035

# Comments on the fossils used to calibrate the phylogenetic trees

## 1.1 Total evidence phylogenetic analysis

The prior assigned to root age relied on the following fossils: the oldest Rodentia, the oldest Aplodontiidae, the oldest Gliridae and the oldest Myodonta (see also *Table 4*). The oldest rodent known to date is *Acritoparamys atavus* (Reithroparamyinae, Ischyromidae) from the latest Paleocene (early Clarkforkian, 56.0 – 55.8 Ma) of Montana (*Janis et al., 2008*; *Korth, 1994*). *Spurimus selbyi* is the oldest aplodontid (*Flynn and Jacobs, 2008*; *Korth, 1994*), and *Eogliravus wildi*, the earliest glirid (*Daams and Bruijn, 1995*; *Dawson, 2003*; *Hartenberger , 1971*). *Spurimus* is a basal member of the aplodontiids, the sister group of sciurids (*Fabre et al., 2012*; *Huchon et al., 2002*), dating back to the middle Eocene (early Uintan, 46.3 – 45.7 Ma) of Wyoming (*Flynn and Jacobs, 2008*). *Eogliravus wildi* is a stem glirid known from isolated cheek teeth from France (*Hartenberger J-L and Hartenberger, 1971*) and an exceptionally preserved specimen (articulated skeleton with pelage impression) from the Messel *Lagerstätte* in Germany (*Storch and Seiffert, 2007*). *Eogliravus* and *Gliravus* are distinguished from all other glirids by their plesiomorphic protrogomorphous zygomasseteric anatomy which they share with basal rodents (*Daams and Bruijn, 1995*; *Storch and Seiffert, 2007*; *Vianey-Liaud, 1985*). The French fossils date back to the earliest Eocene (biozone MP10) (*Daams and Bruijn, 1995*; *Dawson, 2003*), yielding an age range of 50.7 – 47.4 Ma (age boundaries after *Vanderberghe et al., 2012*). Finally, *Erlianomys combinatus*, recovered at the lower part of the Arshanto Fromation of Inner Mongolia (China) (*Li and Meng, 2010*; *Wu et al., 2012*), represents the earliest record of myodonts, the mouse-like rodent clade (*Blanga-Kanfi et al., 2009*; *Fabre et al., 2012*; *Huchon et al., 2002*) (sister group to Aplodontiidae + Sciuridae + Gliridae; *Blanga-Kanfi et al., 2009*; *Fabre et al., 2012*; *Huchon et al., 2002*). This occurrence has been relatively dated by means of bio- and magnetostratigraphy, yielding an age range of 54.8 – 53.9 Ma (early Eocene) (*Wang et al., 2010*). All these data were considered in the calculation of root age and a uniform prior of 52.7 – 45.7 was finally used.

The oldest record of *Douglassciurus jeffersoni*, one of the oldest sciurids, is included as tip date. *Douglassciurus jeffersoni* dates back to the late Eocene (middle Chadronian, 36.6 – 35.8 Ma) of Wyoming and is the oldest-known tree squirrel (*Emry and Korth, 1996*; *Goodwin, 2008*), being somewhat younger than the oldest purported flying squirrel, *Hesperopetes thoringtoni* (*Emry and Korth, 2007*). Although it is not a Sciurinae, the postcranial skeleton of *Douglassciurus* is surprisingly similar to that of extant tree squirrels such as *Sciurus* (*Emry and Thorington, 1982*; *Emry and Thorington, 1984*), but the archaic protrogomorphous zygomasseteric structure of the skull (*Emry and Korth, 1996*; *Emry and Thorington, 1982*) is shared with aplodontiids and other primitive rodent groups such as ischyromids. Consequently, *Douglassciurus* is generally considered the most basal sciurid, since all other extinct genera, including other stem taxa are sciuromorphous or show a modified protrogomorphous zygomasseteric structure (*Black, 1963*; *Goodwin, 2008*; *Korth and Emry, 1991*). This basal position is confirmed by our results (see *Figure 7*). Finally, the record of *Miopetaurista neogrivensis* from ACM/C5-D1 (11.63 Ma) is also considered as tip date in this analysis.

## 1.2 Node dating phylogenetic analysis

Node dating for the Sciurinae (*Figure 7—figure supplement 1*) considered two different calibration points: one for the divergence between *Sciurus* and *Tamiasciurus*, and another between *Petaurista* and the remaining Pteromyina, that is, the large-sized flying squirrel clade

(*Table 4*). The minimum age for the split between *Sciurus* and *Tamiasciurus* is given by the oldest record of the former genus, *Sciurus olsoni* (*Emry et al., 2005*), dating back to the middle–late Miocene (Clarendonian, 13.6 – 10.3 Ma) from the Truckee Formation of Nevada (USA) (*Emry et al., 2005*; *Goodwin, 2008*; *Woodburne, 2004*). *Protosciurus* and *Miosciurus*, which are recorded from the Oligocene of North America, have been considered sciurins (*Black, 1963*; *Goodwin, 2008*), but their archaic zygomasseteric structure indicates that they should be better regarded as stem sciurids. *Miopetaurista neogrivensis* from ACM/C5-D1 (11.6 Ma) provides a minimum age for the divergence of *Petaurista* from the remaining Pteromyina, because *Miopetaurista* is recognized as its sister taxon by total evidence analysis (*Figure 7*).

The soft maximum for the calibration point in the Pteromyina was informed by the most recent age estimate of the earliest Xerinae, the sister group to the Sciurinae (*Mercer and Roth, 2003*; *Steppan et al., 2004*). Xerines are the most diverse subfamily of extant squirrels (*Koprowski et al., 2016*; *Thorington et al., 2012*), including the speciose Holarctic ground squirrels (Marmotini) as well as the African ground (Xerini) and tree squirrels (Protoxerini). Ground squirrels (sensu lato) have a rich fossil record and include some of the oldest-known squirrels, such as *Palaeosciurus goti* from the early Oligocene of France (*Vianey-Liaud, 1974*; *Vianey-Liaud, 1975*), the earliest squirrel to be recorded in Eurasia. A high diversity of ground squirrels is known from the late Oligocene to the middle Miocene of North America, including *Nototamias*, *Palaearctomys* and *Protospermophilus*, among other genera (*Goodwin, 2008*; *Korth, 1994*). While some authors have included these genera in the marmotin tribe (*Black, 1963*; *Korth, 1994*), recent revisions regard them as stem ground squirrels of unknown affinities (*Goodwin, 2008*), because they lack the diagnostic features characterizing extant subfamilies. The same can be argued about the genus *Palaeosciurus*, which is recognized as a 'ground squirrel' on the basis of limb proportions (*Vianey-Liaud, 1974*; *Vianey-Liaud, 1975*), but does not show the diagnostic features of extant groups. The genus *Miospermophilus* is tentatively included into the marmotini by some authors (*Goodwin, 2008*; *Korth, 1994*), which (if correct) would push the record of the subfamily as far back as 30.0–28.0 Ma (early Arikarean, late Oligocene) (*Goodwin, 2008*). However, the skull of this genus, which would unambiguously show if it is a marmotin or not, is unknown. Xerins are the most basal tribe of xerines and include just six extant species. Their oldest representative is *Heteroxerus*, which is first recorded in the late Oligocene (Chattian, MP28) of Spain (*Álvarez-Sierra et al., 1999*), with an estimated age of 26.4–25.0 Ma (*Barberà et al., 2001*). This genus is abundantly represented in the European Miocene by mandibular and cranial fragments as well as isolated cheek teeth (*De Bruijn, 1999*). The soft maximum for this calibration point is also informed by the minimum age of the oldest estimate of *D. jeffersoni*, basal to crown Sciuridae (*Table 4*). For the calibration point in the sciurins, the soft maximum was informed by the age of *M. neogrivensis* and the ages of oldest *Heteroxerus* and *D. jeffersoni* stated above. The minimum age of the oldest aplodontiid (45.7 Ma) is taken as the maximum possible age of both calibration points.

## Appendix 2

DOI: https://doi.org/10.7554/eLife.39270.036

# Morphological characters for phylogenetic analyses

Phylogenetic analyses consider 36 different species including representatives of all extant flying squirrel genera but *Biswamayopterus*, which is known by very few specimens (*Koprowski et al., 2016*; *Thorington et al., 2012*). In addition, species of the tree squirrels *Ratufa*, *Sciurus*, *Callosciurus* and *Protoxerus* are included. Finally the outgroups include *D. jeffersoni*, the oldest-known squirrel for which well-preserved cranial and postcranial material is available (*Emry and Korth, 1996*; *Emry and Thorington, 1982*), and *Aplodontia rufa*. The latter is the only living aplodontiid, which has been long recognized as the sister group of sciurids (*Simpson, 1945*), as confirmed by recent molecular phylogenetic analyses (*Adkins et al., 2001*; *Fabre et al., 2012*; *Huchon et al., 2002*; *Montgelard et al., 2008*; *Montgelard et al., 2002*). The morphological character matrix comprises 105 characters: 45 dental (1–45), 26 cranial (46–71), 31 postcranial (75–105) and three characters related to the muscles that support the patagium (72–74). Dental and cranial characters are based on *Thorington et al., 2002* with ten new characters added in order to resolve the relationships of *A. rufa* and *D. jeffersoni* to other taxa. Most of these new characters are taken from *Korth and Emry, 1991*. The inclusion of *A. rufa* as outgroup implied some modifications and additions in character states of the matrix of *Thorington et al., 2002*. The postcranial characters are mainly based in *Thorington et al., 2005*. For *D. jeffersoni* the character list from *Thorington et al., 2002* was completed with data from *Emry and Korth, 1996*. Dental and cranial character states for most extant taxa were directly recorded after the study of the material housed within the collections of the Naturalis Biodiversity Center of Leiden (*Aeromys tephromelas; Aplodontia rufa; Belomys pearsonii; Callosciurus notatus; Eupetaurus cinereus; Glaucomys sabrinus; Glaucomys volans; Hylopetes alboniger; Hylopetes phayrei; Hylopetes sagitta; Iomys horsfieldii; Petaurista alborufus; Petaurista elegans; Petaurista petaurista; Petaurista philippensis; Petinomys setosus; Protoxerus stangeri; Pteromys volans; Ratufa bicolor; Ratufa affinis; Sciurus carolinensis; Sciurus vulgaris*), the Muséum National d'Histoire Naturelle of Paris (*Iomys horsfieldii, Petaurillus kinlochii, Pteromyscus pulverulentus, Trogopterus xantiphes*), and the Institut Català de Paleontologia Miquel Crusafont (*Callosciurus notatus; Glaucomys sabrinus*). Similarly, postcranial character states were directly recorded for *Aplodontia rufa, Callosciurus notatus, Glaucomys sabrinus, Glaucomys volans, Hylopetes sagitta, Petaurista petaurista, Ratufa affinis, Sciurus carolinensis* and *Sciurus vulgaris*. Characters on the muscles that support the patagium and characters for all other extant species in the analyses were taken from the literature. Postcranial material is generally rare in museum collections of extant rodents, and generally only pelts and skulls are available for most species. For certain genera such as *Petaurista* or *Hylopetes* only the postcranial material of a single species could be studied. Therefore, for most flying squirrel species character states are based in *Thorington et al., 2005*. We have assumed that postcranial character states are the same for all the species within a given genus, which is apparently reasonable since the diagnoses for most squirrel genera and species take into account cranial and dental traits (*Ellerman, 1940*; *Moore, 1959*). Below we list the morphological characters used in the analyses. Character treatment is indicated with the abbreviations 'O' for ordered characters and 'U' for unordered characters. See *Supplementary file 3* for the complete morphological character matrix.

1. Hypselodont cheek teeth (O): (0) absent; (1) present.
2. P3 (O): (0) absent; (1) very small to small; (2) intermediate to large.
3. P4 size relative to M1 size (O): (0) smaller; (1) approximately equal; (2) larger.
4. Crenulation in the upper cheek teeth (O): (0) none; (1) low; (2) heavy.
5. Enamel surface in the upper cheek teeth (U): (0) smooth; (1) folds: lumpy appearance; (2) pitting; (3) well-defined lophules.

6. Color of the upper incisors (U): (0) deep orange; (1) yellowish orange; (2) yellow; (3) variable in color.
7. Longitudinal lophs in the upper cheek teeth (O): (0) none; (1) one; (2) two or more.
8. Hypocone (U): (0) absent; (1) protocone and hypocone separated from each other by a small groove; (2) large protocone connected by a thin loph tow a smaller hypocone, but both cusps moderately separated; (3) large hypocone well separated from the protocone.
9. Mesostyle on the P4 (U): (0) absent; (1) variable; (2) present; (3) present and very broad.
10. Mesostyle on the M1 and M2 (U): (0) absent; (1) variable; (2) present; (3) present and very broad.
11. Mesostyle on the M3 (U): (0) absent; (1) variable; (2) present; (3) present and very broad.
12. Mesostyle position in P4 (U): (0) N/A; (1) connecting with the paracone; (2) connecting with the metacone; (3) isolated.
13. Mesostyle position on the M1/M2 (U): (0) N/A; (1) connecting with the paracone; (2) connecting with the metacone; (3) isolated.
14. Mesostyle position on the M3 (U): (0) N/A; (1) connecting with the paracone; (2) connecting with the metacone; (3) isolated.
15. Mesoloph on the P4 (O): (0) obsolete; (1) short; (2) long.
16. Mesoloph on the M1/M2 (O): (0) obsolete; (1) short; (2) long.
17. Mesoloph on the M3 (O): (0) obsolete; (1) short; (2) long.
18. Divided parastyle on the P4 (O): (0) absent; (1) present.
19. Divided parastyle on the M1/M2 (O): (0) absent; (1) present.
20. Protoconule on the P4 (O): (0) not evident; (1) incorporated into the protoloph but evident; (3) pinched off from the protoloph.
21. Protoconule on the M1/M2 (O): (0) not evident; (1) incorporated into the protoloph but evident; (3) pinched off from the protoloph.
22. Protoconule on the M3 (O): (0) not evident; (1) incorporated into the protoloph but evident; (3) pinched off from the protoloph.
23. Metaconule on the P4 (O): (0) not evident; (1) incorporated into the metaloph but evident; (3) pinched off from the metaloph.
24. Metaconule on the M1/M2 (O): (0) not evident; (1) incorporated into the metaloph but evident; (3) pinched off from the metaloph.
25. Metaconule on the M3 (U): (0) not evident; (1) incorporated into the metaloph but evident; (3) pinched off from the metaloph.
26. Metaloph connection on the P4 (U): (0) absent; (1) connected to the protocone; (2) connected to the hypocone.
27. Metaloph connection on the M1/M2 (U): (0) absent; (1) connected to the protocone; (2) connected to the hypocone.
28. Metaloph connection on the M3 (U): (0) absent; (1) connected to the protocone; (2) connected to the hypocone.
29. Protoloph-protocone connection on the P4 (U): (0) absent; (1) weak; (2) variable; (3) strong.
30. Protoloph-protocone connection on the M1/M2 (U): (0) absent; (1) weak; (2) variable; (3) strong.
31. Protoloph-protocone connection on the M3 (U): (0) absent; (1) weak; (2) variable; (3) strong.
32. Metaloph connection to protocone/hypocone on P4 (U): (0) N/A; (1) weak; (2) variable; (3) strong.
33. Metaloph connection to protocone/hypocone on M1/M2 (U): (0) N/A; (1) weak; (2) variable; (3) strong.
34. Metaloph connection to protocone/hypocone on M3 (U): (0) N/A; (1) weak; (2) variable; (3) strong.
35. Distolingual flexus (O): (0) absent; (1) present.
36. Lingual cingulum (O): (0) absent; (1) present.
37. Metacone on M3 (O): (0) absent; (1) present.
38. Hypocone on M3 (O): (0) absent; (1) present.
39. Anterolophid and metalophid development in the lower molars (U): (0) anterolophid and metalophid absent; (1) anterolophid and metalophid complete and joining the metaconid;

(2) metalophid joining anterolophid before reaching the metaconid; (3) anterolophid reaching the metaconid and protolophid terminating in the talonid basin.

40. Mesostylid development in all lower cheek teeth (O): (0) absent; (1) present; (2) present and very broad.
41. Anterosinusid (O): (0) obsolete; (1) present.
42. Hypoconulid on the p4 (O): (0) obsolete; (1) present.
43. Hypoconulid on the m1/m2 (O): (0) obsolete; (1) present.
44. Hypoconulid on the m3 (O): (0) obsolete; (1) present.
45. Complexity of the lower cheek teeth pattern (O): (0) simple; (1) somewhat complex; (2) complex.
46. Zygomasseteric structure (O): (0) protrogomorphous; (1) sciuromorphous.
47. Orientation of the zygomatic plate (O): (0) completely horizontal; (1) rising only slightly above the level of the cheek teeth; (2) anterodrosally tilted.
48. Shape of the infraorbital foramen (O): (0) round; (1) laterally compressed.
49. Position of the infraorbital foramen (O): (0) high; (2) low.
50. Orientation of angular process in the mandible (O): (0) parallel to the rest of mandibular processes; (1) bent medially.
51. End of masseteric fossa in the mandible (O): (0) below m1; (1) below p4.
52. Pterygoid ridges (O): (0) short; (1) elongated.
53. Basioccipital orientation (O): (0) vertical; (1) tilted ventrally.
54. Length of the rostrum (O): (0) short; (1) intermediate; (2) long.
55. Width of the rostrum (O): (0) wide; (1) intermediate; (2) narrow.
56. Length of incisive foramina (O): (0) short; (1) intermediate; (2) long.
57. Masseteric tubercles (O): (0) none; (1) not prominent; (2) somewhat prominent; (3) prominent.
58. Shape of masseteric tubercles (O): (0) N/A; (1) flat; (2) bulbous; (3) bulbous and tine-like.
59. Openness of the distal half of the orbit (O): (0) mostly open; (1) somewhat open; (2) slightly open.
60. Postorbital process of the frontal (O): (0) absent; (1) short; (2) long.
61. Dorsal process of the jugal (O): (0) absent; (1) short; (2) long.
62. Orientation of the pterygoids (O): (0) parallel; (1) pinched in.
63. Zygomaticomandibularis fossa (O): (0) deep, (1) shallow; (2) obsolete
64. Stapedial artery (O): (0) enclosed within a bony tube; (1) no stapedial artery (i.e., lack of stapedial and sphenofrontal foramina).
65. Shape of the auditory bulla (O): (0) flask-shaped; (1) oval.
66. Number of septa per bulla (U): (0) zero to one; (1) two to three; (2) more than three, not arranged in a honeycomb pattern; (3) more than three and arranged in a honeycomb pattern; (4) more than three, arranged as a sliver.
67. Width of the bulla (O): (0) narrow; (1) intermediate; (2) wide.
68. Inflation of the bulla (O): (0) inflated; (1) flat.
69. Distance between bullae (O): (0) narrow; (1) intermediate; (2) wide.
70. Size of the bony process from the opening of ear canal (O): (0) small; (1) large.
71. Mastoid process inflated (O): (0) no; (1) yes.
72. Origin of semitendinosus III muscle (O): (0) inseparable from origin of semitendinosus II; (1) caudal vertebrae 5–8; (2) caudal vertebrae 9–15.
73. Semitendinosus III discrete or joined (O): (0) inseparable from semitendinosus I and II; (1) muscular attachment to semitendinosus I, II; (2) tendinous attachment to semitendinosus I, II; (3) separate, but lies just caudal to the other heads of semitendinosus; (4) well separated from the rest of musculature, deeply embedded in the skin of the uropatagium.
74. Insertion of the tibiocarpalis muscle (U): (0) absent; (1) distal tibial tuberosity; (2) metatarsal I; (3) metatarsal II.
75. Elevated scapholunate tuberosity on the pisiform (O): (0) absent; (1) present.
76. Triquetral process on the pisiform (O): (0) absent; (1) present.
77. Twisting of the pisiform (O): (0) yes; (1) no.
78. Enlarged surface for pisiform-triquetrum articulation on the ulnar flange (O): obsolete; (1) present, not prominent; (2) present, prominent.
79. Enlarged surface for pisiform-triquetrum articulation on the radial flange (O): obsolete; (1) present, not prominent; (2) present, prominent.
80. Articulation between metacarpal II and central (O): (0) yes; (1) no.

81. Scapholunate-capitate articulation (O): (0) no; (1) yes.
82. Centrale-hamate articulation (O): (0) no; (1) yes.
83. Lesser multangular-scapholunate articulation (O): (0) no; (1) yes.
84. Size of the triquetral facet on the scapholunate (O): (0) short; (1) long.
85. Shape of the pisiform facet on scapholunate (O): (0) circular; (1) grooved.
86. Proximal falciform process on scapholunate (U): (0) obsolete; (1) present, prominent; (2) present, not prominent.
87. Distal falciform process on scapholunate (U): (0) obsolete; (1) present, prominent; (2) present, not prominent.
88. Angle between proximal falciform process and body of the scapholunate (O): (0) perpendicular; (1) acutely angular.
89. Distal tibial tuberosity (O): (0) absent; (1) present.
90. Patagium (O): (0) absent; (1) present.
91. Fused scapholunate (O): (0) no; (1) yes.
92. Deltoid ridge on humerus (U): (0) very broad and prominent; (1) broad to moderately broad; (2) very narrow and knife-like.
93. Lateral supracondylar ridge on humerus (U): (0) large, extending to the lateral edge of the epicondyle; (1) reduced to greatly reduced, not extending to the lateral edge of the epicondyle.
94. Medial epicondyle of the humerus (U): (0) elongated; (1) reduced.
95. Radio-ulnar joint (O): (0) moveable (connected by ligaments or diarthrosis); (1) fused by a prominent syndesmosis.
96. Process that supports the distal compartment for muscle abductor pollicis longus in the distal radius (U): (0) long; (1) short.
97. Olecranon process of ulna (U): (0) long; (1) short.
98. Distal end of ulna (U): (0) robust and broad; (1) delicate and narrow.
99. Styliform process of ulna (U): (0) similar in size to styliform process of radius; (1) more prominent than styliform process in radius.
100. Third trochanter of femur (U): (0) prominent (may be ridge-like); (1) moderately to highly reduced.
101. Position of the third trochanter of femur (U): (0) distal to the lesser trochanter; (1) opposite and proximal to the lesser trochanter.
102. Popliteal fossa of tibia (U): (0) deep; (1) intermediate; (2) shallow.
103. Anterior (cnemial) crest of the tibia (U): (0) prominent; (1) intermediate; (2) faint.
104. Pit on the plantar surface of the astragalus between the calcaneal articular surfaces (U): (0) present; (1) absent.
105. Position of the sustentacular process of the calcaneus (U): (0) opposite to the peroneal process; (1) slightly distal to the peroneal process; (2) clearly more distal to the peroneal process.

## Appendix 3

DOI: https://doi.org/10.7554/eLife.39270.037

# Detailed morphological descriptions and comparisons

## 3.1 Cheek tooth morphology and species attribution

IPS56468. The cheek teeth of the associated skull (*Figure 5—figure supplement 1h-i*) are moderately worn, so certain details cannot be appreciated, especially in the lower series. The enamel in the upper cheek teeth is smooth and the main transverse ridges are thick and simple. The P3 is small and rounded, presenting a single anterolingual cusp. The P4 shows a well-developed parastyle forming a broad platform that makes this premolar longer that wide (see *Supplementary file 4*). The parastyle is as developed as the main labial cusps (paracone and metacone). The protoloph and the metaloph in the P4–M2 are higher than the remaining transverse ridges and parallel, reaching the protocone separately. The protoloph shows a thickening near its lingual end, presumably corresponding to the protoconule, in the P4 and M3, but this conule is missing in the M1 and M2. The metaconule is absent in all upper cheek teeth. The metaloph shows a short distal spur in the P4, whereas a similarly-developed mesial spur is present in the protoloph of the M3. The protocone is longitudinally elongated and continuous with the ectoloph, whereas the hypocone is highly reduced and cannot be distinguished from this ridge. The mesostyle is present in the P4 and M3 but not in the M1 and M2. The M3 is relatively short and shows only two main cusps, protocone and paracone, and a wide valley delimited by a continuous ridge. This molar presents a short, thin and arched metaloph which forms a reduced distolingual valley. This valley is closed by a small style connected to the metaloph by means of a short spur.

In contrast to the upper cheek teeth, the enamel in the lower ones shows extensive meandriform crenulations that mostly affect the valleys but also extend to the main ridges. There are three well-defined cusps (protoconid, metaconid and hypoconid) and three smaller ones (entoconid, mesoconid and mesostylid) surrounding a relatively shallow central valley. The mesostylid is placed quite mesially, next to the metaconid. The hypoconulid is absent. The ectolophid is M-shaped and the sinusid reduced and open. The anterosinusid is highly reduced and closed by an anterior cingulid descending from the anterolophid. The metalophid is short in the p4–m2, not reaching the metaconid. The p4 is markedly longer and narrower than the molars and presents a small and rounded anteroconulid. It shows a short, distally-directed mesolophid placed just distal to the protoconid. The entolophid is thick but it is interrupted before reaching the entoconid. In contrast, the entolophid is thick and complete in the m1–m3. In these molars the entolophid is transverse, merging the entoconid with the ectolophid between the mesoconid and the hypoconid. In addition, in the m1 the mesostylid shows a labially-directed ridge pointing towards an opposite ridge that departs from the ectolophid. These two ridges are interpreted as an incomplete mesolophid and cannot be discerned in the m2. The m3 shows a complex morphology, with a low metalophid that bifurcates into two lower meandriform ridges that join the mesostylid and the mesoconid, respectively. A vestigial mesolophid is also present in the m3 between the mesoconid and the protoconid.

Additional material. Additional remains from ACM/C5-D1 include a maxillary fragment with the broken P4–M2 (IPS43724), two partial mandibles with p4–m3 (IPS43677, IPS85340), 37 isolated cheek teeth and additional cheek teeth fragments (see *Table 2* and *Supplementary file 4*). In addition, a partial cranium (IPS88677) has been recovered in the approximately stratigraphically equivalent site ACM/C8-Af (*Alba et al., 2017*) (*Figure 6—figure supplement 2*). A highly fragmentary third cranium (IPS85410) comes from locality ACM/C6-A5, with an estimated age of also 11.6 Ma (*Alba et al., 2017*). IPS85410 includes only part of the frontal, parietal and maxillary with damaged P4–M3. Finally, another partial mandible with p4–m3 (IPS87560) has been recovered in sector ACM/C8-B (*Figure 3f–g*). Age range for the localities in this sector of the ACM series is from 11.6 to 11.7 Ma (*Alba et al.,*

**2017**), so the age of all additional specimens is very close to that of the partial skeleton IPS56468.

Overall the size and morphology of these additional specimens is very similar to that of IPS56468. However, this additional material shows that some occlusal features are variable, such as the development of the metalophid, the mesolophid and the entolophid in the lower cheek teeth. The metalophid is generally short and does not reach the metaconid, however, this ridge is complete in one isolated p4 (IPS77873) and two m1/m2 (IPS43675, IPS77879). The entolophid can be partial in some premolars, but this ridge is always thick and complete in the molars. A well-defined short to medium-length mesolophid can be recognized in several individuals, such as in IPS43677. However, this ridge (if present) is usually too low to be clearly distinguished from the abundant crenulations that occur in the central valley. Three isolated dp4 have been recovered. These are much smaller but morphologically similar to the p4, although the anteroconulid is more reduced and situated closer to the protoconid. As far as the upper cheek teeth are concerned, the P4 is more variable. A metaconule is generally present, but the protoconule is distinct only in a few specimens. The number of distally-directed spurs in the metalophule varies between two and three, with one specimen (IPS43675) showing a mesially-directed spur as well. The parastyle of the P4 is invariably defined by a single cusp, and a mesostyle is present in all instances. The M1 and M2 do not show conules in the protoloph and the metaloph, but the later ridge may develop a short distally-directed spur as in the P4. The recovered isolated M3 show a well-defined protoconule, a small mesostyle and a small distolingual flexus. Even though the enamel is smooth in the upper cheek teeth, a few specimens (IPS77859, IPS77862, IPS77863) present weak crenulations in the protoloph and metaloph.

Discussion. The diagnoses of fossil 'flying' squirrel genera and species rely entirely on cheek tooth morphology and dimensions. The partial skeleton IPS56468 and additional material described here (**Table 2**) are attributed to the genus *Miopetaurista*. In the ACM localities a second genus of 'flying' squirrel, *Albanensia*, is recorded, but *Miopetaurista* is clearly distinguished by its larger size, absence of enamel crenulations in the upper cheek teeth, and protoloph and metaloph of the P4–M2 parallel instead of lingually convergent (**Daxner-Höck and Mein, 1975**; **De Bruijn, 1999**; **Mein, 1970**). In addition, in *Miopetaurista* the protoloph and metaloph rarely present conules, the hypoconulid is not apparent in the lower cheek teeth, and the anterosinusid is always present, although reduced (**Daxner-Höck and Mein, 1975**; **De Bruijn, 1999**; **Mein, 1970**).

*Miopetaurista* ranges from the early Miocene to the Pliocene of Eurasia and it includes up to ten different species, although some of them are poorly known and the validity of others is questionable (**Casanovas-Vilar et al., 2015**; **De Bruijn, 1999**; **De Bruijn, 1995**). The specimens described here are outstanding because of their large size (**Supplementary file 4**), clearly above the size range of most species of the genus, particularly the older ones such as *Miopetaurista diescalidus*, *Miopetaurista dehmi* and *Miopetaurista lappi*. They further differ from these species in several morphological features including the absence of mesoloph in the upper cheek teeth, the different shape of the P4, and the presence of a complete entolophid in the lower cheek teeth. The roughly coeval *Miopetaurista gaillardi* is also smaller, but shows a similar morphology, including the presence of a complete entolophid in some lower molars. However, this species is distinguished by its shorter P4 as result of the less developed parastyle. *Miopetaurista gibberosa*, the type species of the genus, is only known from a single mandible from the middle Miocene (MN5) of Göriach (Austria) (**Daxner-Höck and Höck, 2015**). The lower cheek teeth of this species are smaller than the described material and in addition they show a highly reduced entolophid. The size of the described specimens is only comparable to those of *Miopetaurista neogrivensis*, *Miopetaurista crusafonti* and *Miopetaurista thaleri*. The latter species is slightly larger and further displays a number of morphological differences (**Mein, 1970**), such as the presence of a complete mesolophid that joins the entoconid in the lower molars. The described material is well above the size range of *M. crusafonti*, which is recorded in other Vallès-Penedès sites of similar age (**Casanovas-Vilar et al., 2015**; **Casanovas-Vilar et al., 2010**). However, the described molars always show a complete entolophid, whereas this ridge is always partial in *M. crusafonti*. Furthermore, in

the studied material the mesolophids are better developed than in *M. crusafonti* and the P4 is less triangular (*Casanovas-Vilar et al., 2015*). The described specimens fit perfectly in size and morphology with *M. neogrivensis*, which is present elsewhere in the Vallès-Penedès Basin (*Aldana Carrasco, 1991*; *Aldana Carrasco, 1992*; *Casanovas-Vilar et al., 2015*). This species is only surpassed in size by *M. thaleri* and shows a more complex occlusal morphology than *M. crusafonti*.

Mein, 1970 described a number of dental features in extant and fossil 'flying' squirrels that were used to distinguish three different groups mainly on the basis of enamel wrinkling and the presence of additional lophules. According to this author *Miopetaurista* (= *Cryptopterus* in Mein (1970) and *Daxner-Höck and Mein, 1975*) would belong to the second group, characterized by its smooth enamel and absence of lophules. This group would also include the extant genera *Glaucomys*, *Eoglaucomys* and *Iomys*. Other common characters of the group are the absence of metaloph on M3, the absence of mesostyle, the presence of anterosinusid in the lower molars and the absence of anteroconid. Although the mesostyle is generally absent in the upper molars of *Miopetaurista*, it frequently occurs in the P4. On the other hand, the anteroconid is absent from the lower molars but present in the p4 and dp4. These dental characters have proven to be inadequate to distinguish between clades of flying squirrels, or even to separate flying squirrels from other squirrels (*Thorington et al., 2002*; *Thorington et al., 2005*). *Petaurista* is characterized by its smooth enamel but shows a much more complex morphology, with additional longitudinal lophules and would belong to a completely different group than *Miopetaurista* according to dental classification (*Mein, 1970*). This contradicts our phylogenetic results (see main text). Nevertheless, there is a remarkable dental similarity between *Miopetaurista* and *Petaurista*, namely the presence of a distolingual flexus. This occurs in all upper cheek teeth in *Petaurista*, whereas in *Miopetaurista* it only occurs in the M3. Such feature is only observed in two other genera from the subtribe Pteromyina, *Aeretes* and *Eupetaurus*, being absent in the Glaucomyina.

## 3.2 Skull morphology

The reconstruction of skull morphology is based on the almost complete cranium and mandible of IPS56468 from ACM/C5-D1 (*Figure 3a–e*, *Figure 6—figure supplement 1*), as well as on the additional information provided by the partial cranium IPS88677 from ACM/C8-Af (*Figure 6—figure supplement 2*) and the partial mandible IPS87560 from sector ACM/C8-B (*Figure 3f–g*). The cranium IPS56468 is laterally crushed in an oblique angle, with the rostrum deviated towards the left side. The zygomatic arch is complete but crushed, with the jugal and other fragments pressed against the orbit. On the other hand, IPS88677 is dorsoventraly crushed and most of the premaxillary and frontal bones are not preserved. The virtual reconstruction of the skull (see Materials and Methods; *Figure 6* and *Video 3*) relied primarily on IPS56468h, but used some elements from IPS88677 when the bone was particularly damaged or missing in the former specimen (zygomatic plate, jugal, pterygoid, alisphenoid; *Figure 6—figure supplement 1-2*, *Table 3*). The description of cranial morphology relies on the original specimens but also considers aspects of the virtual reconstruction concerning overall shape and proportions of certain structures (i.e., rostrum breadth and height, separation between bullae). The skull was directly compared to that of extant flying squirrels housed within the collections of the Naturalis Biodiversity Center of Leiden (*Aeromys tephromelas*; *Belomys pearsonii*; *Eupetaurus cinereus*; *Glaucomys sabrinus*; *Glaucomys volans*; *Hylopetes alboniger*; *Hylopetes phayrei*; *Hylopetes platyurus*; *Hylopetes sagitta*; *Iomys horsfieldii*; *Petaurista alborufus*; *Petaurista elegans*; *Petaurista petaurista*; *Petaurista philippensis*; *Petinomys hageni*; *Petinomys setosus*; *Petinomys vordermanni*; *Pteromys momonga*; *Pteromys volans*) and the Muséum National d'Histoire Naturelle of Paris (*Iomys horsfieldii*, *Petaurillus kinlochii*, *Pteromyscus pulverulentus*, *Trogopterus xantiphes*). Only *Eoglaucomys* and the poorly-known genera *Aeretes* and *Biswamayopterus* could not be directly compared. Comparisons with *Aeretes melanopterus* are based on published data (*Thorington et al., 2002*; *Thorington et al., 2012*). In addition, the posterior region of the skull of *P. petaurista* was scanned in order to compare its inner ear morphology to that of *M.

*neogrivensis* (see Materials and Methods, **Video 4**). Several cranial and mandibular measurements were taken in the specimens of *M. neogrivensis* and some comparative material (see **Supplementary file 5–6**). Measurements were also taken from the virtually reconstructed cranium of *M. neogrivensis* using Avizo 7.0.

Cranium. The cranium is relatively short and wide, as in most large-sized flying squirrels but *Eupetaurus* and a few species of *Petaurista*, such as *Petaurista alborufus* (**Figure 6**). The upper incisors are orthodont, as in most flying squirrels except *Pteromys* and *Aeretes*, which show opisthodont incisors. The incisors are not as wide and robust as in other large-sized flying squirrels. The rostrum is similar in length to that of *Petaurista*, *Aeretes* and *Aeromys*. It is also relatively wide and high, although not as much as in *Aeretes*. The nasals are not preserved in any of the specimens, but their suture with the premaxilla is almost complete in IPS88677 (**Figure 6—figure supplement 2**), showing that they were most likely wide, resembling those in *Aeretes* and *Petaurista*. The incisive foramina are slightly longer than in most flying squirrels, approaching the condition seen in *Aeromys* and *Eupetaurus*. The masseteric tubercles are somewhat prominent and bulbous, being very similar in size and shape to those of *Petaurista*. In addition, there is a prominent semicircular scar for the origin of the superficial masseter (**Ball and Roth, 1995**; **Thorington and Darrow, 1996**) in front of the cheek tooth row. The palate is not particularly wide, being markedly narrower than in *Glaucomys*, *Pteromys* and most *Hylopetes* and *Petinomys* species. A short posterior nasal spine is present at the suture between the palatines. This is highly variable in flying squirrels, but the spine is generally reduced in *Aeromys*, *Hylopetes* and *Glaucomys*. The pterygoid ridges diverge distally, as in *Aeretes*, instead of being parallel as in most flying squirrels or slightly convergent as in *Pteromys* and *Trogopterus*. The auditory bullae are inflated and similar in size to those of *Petaurista* and *Aeromys* and separated by a wide distance as in *Petaurista* and most *Hylopetes* species. This is in marked contrast with *Aeromys* and, especially, *Aeretes*, which show a narrower space between the bullae. A µCT scan of the middle ear cavity of IPS88677 shows the presence of two ventral transbullar septa (**Video 4**), while this feature is not preserved in IPS56468h. Septated bullae occur in the tympanic cavity of certain rodents (Aplodontiidae, Sciuridae, Geomyidae and some Arvicolinae), although their function is unclear (**Mason, 2015**). Several authors have stressed the taxonomic importance of the presence or absence of these structures and the number of septa (**Moore, 1959**; **Pfaff et al., 2015**; **Thorington et al., 2002**). Most squirrels show transbullar septa, and their number typically ranges from one to three, but some flying squirrels show many more, sometimes even arranged in a honeycomb (as in *Petinomys*) or in a sliver pattern (as in *Trogopterus* and *Pteromyscus*) (**Moore, 1959**; **Pfaff et al., 2015**; **Thorington et al., 2002**). *Miopetaurista neogrivensis* shows just two septa, as in most flying squirrels including *Petaurista* (**Video 4**). The external auditory meatus is large and rounded. The bony process from the opening ear canal is large. As in all flying squirrels but *Iomys* and *Petaurillus*, the mastoid process is not inflated.

In dorsal view the postorbital process of the frontal is robust and long, partly enclosing the distal half of the orbit, thus approaching the morphology of large-sized flying squirrels such as *Petaurista* and *Aeromys*. Small-sized flying squirrels usually show more slender and shorter postorbital processes. The frontal is depressed on its anterior part. There is a conspicuous semicircular notch just in front of the postorbital processes. This notch is also present in many flying squirrels, including *Aeromys* and *Petaurista*, but it is more pronounced in *Miopetaurista*, thus approaching the morphology of *Pteromys*, *Glaucomys* and most *Hylopetes* species. The zygomatic plate of the premaxillary bone is similar in size and morphology to that of most flying squirrels, except for *Pteromys* and *Petaurillus*, which show wider plates bordered by more marked ridges. The infraorbital foramen is relatively wide. The zygomatic arches are wide and approximately parallel to the longitudinal axis of the skull. The jugal is deeper than in most flying squirrels and shows a dorsal process pointing towards the postorbital process of the frontal. This process is also observed in many flying squirrel genera, including *Petaurista*, *Aeromys* and *Aeretes*, but it is more pointed than in *Miopetaurista*. The origin of the deep masseter on the jugal bone (**Ball and Roth, 1995**; **Thorington and Darrow, 1996**) defines a marked scar that is larger and extends somewhat more dorsally than in any other studied squirrel. The cranial vault is relatively flat as in most large-sized flying squirrels, except for

*Aeromys tephromelas* and some *Petaurista* species. Small-sized flying squirrels generally show more convex cranial vaults, sometimes markedly inflated as in *Iomys*. The parietal is wide and shows well defined temporal ridges (marking the insertion of the temporalis muscle (**Ball and Roth, 1995**; **Thorington and Darrow, 1996**) that approach each other towards the posterior part of the skull. Even though this occurs to some degree in certain flying squirrels (*Pteromys*, *Hylopetes*), the ridges come so close only in *Eupetaurus*.

Mandible. Both hemimandibles of IPS56468 are damaged in their posterior part, with the mandibular process partially broken and displaced (**Figure 3a–e**). The articular process is completely missing in the left hemimandible (IPS56468j), which also lacks the tip of the coronoid process. On the contrary, both processes are preserved in the right hemimandible (IPS56468i), although detached and moved from their anatomical location. The partial hemimandible IPS87560 preserves the articular process in its original position as well as the angular process (**Figure 3f–g**). A complete mandible was assembled using three-dimensional models of these three specimens digitized by means of photogrammetry (see Materials and Methods; **Figure 1**, **Videos 1** and **3D** model in **Supplementary file 1**). However, the description of mandibular morphology relies on the original specimens, the three-dimensional model being only considered for assessing the shape of the mandibular ramus between the coronoid and articular processes.

The mandible is relatively deep and short, with the lower incisor projecting slightly above the occlusal plane of the cheek teeth. The corpus is strong and deep. The symphysis is strong and the diastema is relatively short. The masseteric ridges converge at the level of the distal root of the m1. The point where the ridges meet does not form a bump or a tubercle as in other sciurids. The masseteric fossa is quite deep. The coronoid process is robust and hook-shaped, resembling that of *Aeretes*, *Aeromys*, *Petaurista*, *Belomys*, *Petinomys* and some *Hylopetes* species such as *Hylopetes sagitta*. The coronoid is also noticeably higher than the articular process as in all these genera, but rises in an angle closer to that of *Hylopetes* and *Petinomys* and not as steep as in *Petaurista*, *Aeretes*, *Belomys* or *Trogopterus*. IPS56468i partially preserves the outline of the ramus between the coronoid and articular processes. These two processes are not as separated as in *Petaurista*, *Aeretes*, *Belomys* or *Trogopterus*, being closer as in *Aeromys*, *Glaucomys* and some species of *Hylopetes* and *Petinomys*. IPS87560 preserves the articular process in its original position and shows that it is long and projects distally. *Aeretes*, *Glaucomys* and *Hylopetes* show similarly long articular processes, while most large-sized flying squirrels such as *Aeromys*, *Eupetaurus* and *Petaurista* show shorter processes. The articular condyle is longitudinally elongated. The angular process is of intermediate length and distally directed, not extending far ventrally. As in *Petaurista*, *Aeromys* and *Hylopetes*, for example, the angular process does not reach farther posteriorly than the articular one. The inner surface of the angular process shows a marked fossa for the insertion of the medial pterygoid partially divided by a marked longitudinal ridge. The ventral edge of the angular process presents a well-defined area for the insertion of the superficial masseter bordered by faint ridges. Overall, mandibular morphology and robusticity recall *Aeromys* and *Hylopetes*, being slightly more gracile and elongated than in *Aeretes* and *Petaurista*, for example.

The skull of *Miopetaurista* resembles that of other large-sized flying squirrels, particularly *Aeromys* and *Petaurista*, which are characterized by their short and wide rostrum, moderately inflated bullae and relatively wide posterior region of the skull. Other details are strikingly similar to these extant genera, including the robust and long postorbital process that partially encloses the orbit, the well-developed jugal process in the zygomatic arch, or the presence of two septa in the tympanic cavity. Most of the smaller flying squirrels show more elongate muzzles, slender or shorter postorbital processes and, in some cases, a higher number of transbullar septa. However, the mandible of *Miopetaurista* is somewhat slenderer than in most large-sized flying squirrels, showing a more elongated articular process resembling that of *Hylopetes* and other small-sized flying squirrels instead of the lower and shorter process of *Petaurista*. The characters of the skull agree with our phylogenetic results (see main text and **Figure 7**) in showing a close relationship between *Miopetaursita* and extant large-sized flying squirrels, particularly *Petaurista*. However, we must note that none of these characters

distinguishes flying squirrels exclusively. Indeed, overall skull morphology of some large-sized tree squirrels such as *Ratufa* and *Protoxerus* (*Ellerman, 1940*; *Moore, 1959*) is similar to that of large-sized flying squirrels.

## 3.3 Postcranial morphology

The partial skeleton IPS56468 comprises up to 80 different elements (*Table 1*) including the skull, most of the long bones of the fore- and hindlimbs, elements of the wrist and ankle, metapodials and phalanges, fragmentary pelves and clavicles, a few vertebrae from different regions and scarce rib fragments. Some of these elements, particularly those from the hindlimb, are remarkably well preserved and were even found in anatomical connection (*Figure 2* and *Figure 2—figure supplement 1*). Here we briefly describe the postcranial skeleton, outlining the distinguishing morphological features of flying squirrels (*Thorington et al., 2005*), which are mostly related to gliding locomotion. The skeletal elements of *M. neogrivensis* were compared with those of extant squirrels (*Supplementary file 7*) in the collections of the Institut Català de Paleontologia Miquel Crusafont (*Glaucomys sabrinus*, *Sciurus niger*) and the Naturalis Biodiversity Center (*Urocitellus parryii*, *Xerus erythropus*, *Callosciurus prevostii*, *Sciurus vulgaris*, *Hylopetes sagitta* and *Petaurista petaurista*).

The recovered long bones include the two humeri and femora, as well as the right tibia. Only the distal half of the left tibia and that of the right radius have been preserved, the distal epiphysis of the radius further being damaged. Complete long bones of flying squirrels are much longer and more slender than those of tree and ground squirrels (*Samuels and Van Valkenburgh, 2008*; *Thorington et al., 2005*; *Thorington and Heaney, 1981*; *Thorington and Santana, 2007*), and this is also the case for *M. neogrivensis* (*Figure 5—figure supplement 1*). Moreover, several additional features can be recognized in the proximal and distal ends of these bones as described below.

Humerus. On the proximal half of the humerus, the deltoid ridge is not as prominent as in tree and specially ground squirrels (*Figure 5—figure supplement 1a-e*). In flying squirrels this ridge is also sharper and much thinner than in ground squirrels and most tree squirrels. The deltoid ridge in *M. neogrivensis* is less prominent than in *Hylopetes*, being roughly comparable to that of *Petaurista*. The reduced deltoid ridge of flying squirrels probably allows increased flexibility at the shoulder joint, as opposed to the condition of tree and ground squirrels, which serves to increase the power of forelimb flexion at the expense of flexibility at this joint (*Thorington et al., 2005*). The distal end of the humerus shows additional differences between the three kinds of squirrels. The epicondylar ridge is large in tree and particularly in ground squirrels, extending to the lateral edge of the epicondyle (*Figure 5—figure supplement 1f-g*). In most flying squirrels, this ridge is reduced and does not reach the epicondyle (*Figure 5—figure supplement 1h-j*). In large-sized flying squirrels and *M. neogrivensis* this ridge is even more reduced than in small-sized flying squirrels such as *Hylopetes* and *Eoglaucomys*. Flying squirrels and *M. neogrivensis* are further characterized by a small medial epicondyle that contrasts with the much larger one of ground squirrels and the elongated epicondyle of tree squirrels (*Figure 5—figure supplement 1f-j*). The reduction of the lateral epicondylar ridge in flying squirrels represents a reduction in the leverage of the extensor muscles for flexion of the elbow joint, thus facilitating elbow joint extension for gliding (*Thorington et al., 2005*). Finally, the trochlea is not sharply angled as in tree and ground squirrels, being flatter as in flying squirrels (*Figure 5—figure supplement 1f-j*).

Radius. The radius is not well preserved, so that the very few features that distinguish the radius of flying squirrel from that of other squirrels cannot be observed.

Carpal bones. Carpals are the most diagnostic elements of pteromyins and can be used to unambiguously diagnose the group because of their role in the extension of the patagium (*Thorington, 1984*; *Thorington et al., 2005*; *Thorington et al., 2002*; *Thorington and Darrow, 2000*). The matrix removed during the preparation of the two blocks containing the skeleton of *M. neogrivensis* was screen-washed, allowing for the recovery of small elements, including the complete right scapholunate and the dorsal end of the pisiform (*Figure 4* and

*Video 2*). The pisiform shows three processes for articulation with the ulna, triquetrum and scapholunate. The presence of a radial thumb-like process that articulates with the scapholunate is a pteromyin synapomorphy not observed in other squirrels. In *M. neogrivensis*, as well as in other flying squirrels, the scapholunate process of the pisiform is well developed and fits into a distinct facet of the former bone (*Figure 4* and *Video 2*). The patagium in flying squirrels is supported by the styliform cartilage, which is strongly bound to the palmar end of the pisiform, and extended and raised by means of the abductor of the thumb on the opposite side of the wrist (*Thorington, 1984*; *Thorington et al., 1998*; *Thorington and Darrow, 2000*). The articulation of the pisiform with both the triquetrum and the scapholunate causes the proximal row of carpal bones to function as a single unit, providing a more stable base for the styliform cartilage and the wing tip during gliding (*Thorington, 1984*; *Thorington et al., 1998*; *Thorington and Darrow, 2000*). On the other hand, the articular surface between the radius and scapholunate is more spherical in flying squirrels and *M. neogrivensis*, whereas it is planoconvex in tree squirrels (*Thorington, 1984*; *Thorington and Darrow, 2000*). Maximum extension of the wing tip of the patagium is achieved when the hand is dorsiflexed and radially deviated (*Thorington, 1984*; *Thorington et al., 1998*; *Thorington and Darrow, 2000*). Dorsiflexion is required to elevate the tip of the patagium at the wrist relative to the rest of the membrane and it would likely function to reduce induced drag when gliding (*Thorington et al., 1998*). The greater convexity and extension of the dorsal articular surface of the scapholunate in the pteromyins allow for a greater angular movement at the proximal wrist joint, which is required to bring the hands to gliding position. Besides the scapholunate and ulnar process, the pisiform of *M. neogrivensis* shows a distinct triquetral process (*Figure 4* and *Video 2*). This process is diagnostic of the Pteromyina (i.e., the large-sized flying squirrel clade, see main text) and fits between the palmar surfaces of the scapholunate and triquetrum (*Thorington et al., 2002*; *Thorington and Darrow, 2000*). Therefore, the proximal carpal bones of *M. neogrivensis* not only unambiguously indicate that it is a flying squirrel, but also allow assigning it to one of the two extant flying squirrel clades (*Figure 7*).

Femur. The proximal end of the femur shows reduced lesser trochanter and third trochanter, as seen in flying squirrels (*Figure 5—figure supplement 1k-o*). The lesser trochanter is larger in most tree squirrels, extending medially as far or farther than the femoral head. In marmotin ground squirrels, the lesser trochanter is even shorter than in flying squirrels (*Figure 5—figure supplement 1k*), but it is more robust and posteromedially directed, being presumably linked to their digging activities and burrowing habits. In flying squirrels, the third trochanter is markedly more reduced than in other squirrels and it is placed more proximally. It has been proposed that the reduction of the third trochanter and the trochanteric ridge in flying squirrels may increase flexibility at the hip joint (*Thorington et al., 2005*), as described for the case of the shoulder joint. Although the distal epiphysis shows fewer diagnostic characters, *M. neogrivensis* resembles large-sized flying squirrels (*Petaurista*, *Aeromys*) in showing a broader patellar groove (*Figure 5f–j*).

Tibia. The popliteal fossa is shallow and not limited by low ridges as in tree and small flying squirrels (*Figure 5—figure supplement 1t-s*). This fossa is generally deeper in tree squirrels, and markedly pronounced in ground squirrels. The anterior crest is less pronounced than in tree and ground squirrels (*Figure 5—figure supplement 1t-s*). The anterior surface of the distal epiphysis does not present the tuberosity for the origin of the tibiocarpalis muscle that characterizes the Glaucomyina (i.e., the small-sized flying squirrel clade, see main text; *Figure 5h–j*; *Thorington et al., 2002*). The tibiocarpalis runs from the ankle to the tip of the styliform cartilage and defines the margin of the flying membrane. In the Pteromyina, and presumably in *M. neogrivensis* as well, this muscle originates from the dorsal surface of the first or second metatarsals (*Thorington et al., 2002*). This further supports the inclusion of *M. neogrivensis* among the latter subtribe.

Astragalus. The trochlear surface of the astragalus is said to be more V-shaped in flying squirrels than in other squirrels (*Thorington et al., 2005*). In the extant sciurids we examined, the shape of the trochlea is similar in tree and flying squirrels, although in small-sized flying squirrels (*Hylopetes* and *Glaucomys*) the angle between the lateral and medial malleolar

surfaces in the trochlea is sharper. The plantar surface of the astragalus shows a well-defined sulcus tali between the posterior and the middle calcaneal articular surfaces (*Figure 2k–l*). This sulcus accommodates the astragalocalcaneal interosseus ligament and synovial tissue of the joints. In flying squirrels the two calcaneal articular surfaces are enlarged and occlude the sulcus to the point that it is barely appreciated (*Ginot et al., 2016*; *Thorington et al., 2005*). This derived morphology has been related to a greater mobility of the ankle joint associated with extreme plantar inversion of the foot during gliding (*Thorington et al., 2005*). A greatly reduced sulcus is also present in the tree squirrels of the tribes Sciurini (*Sciurus*, *Tamiasciurus*) and Protoxerini (*Funisciurus*, *Paraxerus*), presumably permitting the reversal of the foot during headfirst descent of trees (*Ginot et al., 2016*; *Thorington et al., 2005*). On the contrary, this sulcus is really prominent in *Ratufa* (tribe Ratufini) and at least in some Callosciurini (*Callosciurus*, *Sundasciurus*) (*Thorington et al., 2005*). However, all these tree squirrels are perfectly capable of rotating their feet. A similarly developed sulcus is present in *Douglassciurus jeffersoni* and has been interpreted as a plesiomorphic feature unrelated to tree climbing (*Emry and Thorington, 1982*; *Emry and Thorington, 1984*). On the other hand, in marmotine ground squirrels, which do not climb trees, the sulcus is very narrow or completely missing (*Thorington et al., 2005*). *Miopetaurista neogrivensis* shows a deep groove comparable to that of *Ratufa*. Interestingly, the astragalus of the early Miocene North American putative flying squirrel *Petauristodon pattersoni* presents a similarly pronounced sulcus (*Pratt and Morgan, 1989*) which has been previously taken as a proof for the exclusion of this genus from pteromyins (*Thorington et al., 2005*). Clearly, the presence of an occluded sulcus in the plantar surface of the astragalus is not diagnostic of flying squirrels and is not required for foot inversion.

Calcaneus. The sustentacular process of the calcaneus is clearly larger than the fibular process and it is placed slightly more distally (*Figure 2m–n*), as observed in tree and flying squirrels (*Ginot et al., 2016*; *Thorington et al., 2005*). In ground squirrels both processes are aligned opposite to one another and the sustentacular process is only slightly larger than the fibular one, resulting in a cruciform calcaneus. As in flying squirrels and some tree squirrels of the tribes Protoxerini, Ratufini and Callosciurini, *M. neogrivensis* shows a pronounced calcaneal sulcus between the two facets for articulation with the astragalus. The morphology of the calcaneus is diagnostic for ground squirrels but does not appear to be of use in discriminating between tree and flying squirrels.

Vertebrae. Several vertebrae of *M. neogrivensis* have been recovered (*Figure 2g–h*, *Table 1*), including a partial axis, a partial cervical (C3), four partial thoracics (T1?–T4?), four lumbars in anatomical connection (L3l–L5; *Figure 2g–h*) and four caudals that correspond to the mid part of the tail. Small-sized flying squirrels of the genus *Glaucomys* have more elongated lumbar and mid-caudal vertebrae as compared to similarly sized tree squirrels (*Microsciurus*) and show a reduction of the transverse processes in lumbar vertebrae (*Thorington and Santana, 2007*). The lumbar vertebrae of *Petaurista* are also slenderer than in tree squirrels and present reduced transverse processes oriented parallel to the vertebral body. The four recovered lumbar vertebrae of *M. neogrivensis* show the characteristic morphology of *Petaurista*. The elongation of lumbar vertebrae and limbs determines the size and shape of the patagium, which in turn dictates important aerodynamic features such as wing loading and aspect ratio (*Thorington and Heaney, 1981*; *Thorington and Santana, 2007*). The elongation of these elements results in a decrease in wing loading of flying squirrels.

Clavicle. Two clavicles have been recovered, the right one missing only the articulation with the scapula. The clavicle is straight and slender as in arboreal and flying squirrels, while in ground squirrels this bone is conspicuously robust.

The remaining recovered bones (*Table 1*) include a partial pelvis (*Figure 2i–j*), some rib fragments and two sternebrae. In addition, several elements of the ankle and foot have been recovered, including the navicular, two cuneiforms, and several metatarsals, with left metatarsals 2 – 4 in anatomical connection. Most phalanges were recovered after screen-washing and are difficult to assign to ray. Four proximal hand phalanges as well as four proximal foot phalanges have been recovered. Seven intermediate phalanges and several

phalangeal fragments are also attributed to the foot. Metapodials as well as proximal and intermediate phalanges are slender as in tree and flying squirrels. Six distal phalanges (tentatively attributed to the foot) have also been recovered. These are markedly sharp and curved as in most tree and flying squirrels, instead of blunt as in many ground squirrels.

