## [Decision Letter]

Thank you for submitting your article "Oldest skeleton of a fossil flying squirrel casts new light on the origin of the group" for consideration by *eLife*. Your article has been reviewed by three peer reviewers, including Min Zhu as the Reviewing Editor and Reviewer #1, and the evaluation has been overseen by Diethard Tautz as the Senior Editor. The following individuals involved in review of your submission has agreed to reveal their identity: Robert Emry (Reviewer #2).

The reviewers have discussed the reviews with one another and the Reviewing Editor has drafted this decision to help you prepare a revised submission.

This is a well-written, well organized and elegantly illustrated manuscript to address the oldest known near-complete skeleton of a fossil flying squirrel, dating back to around 12 million years ago. The anatomical descriptions and comparisons of this fossil material add to our understanding of the biology and evolutionary history of flying squirrels.

Summary:

Flying squirrels are the only group of gliding mammals with a remarkable diversity and wide geographical range. However, their evolutionary history is still not well known. Thus far, the identification of extinct flying squirrels has been exclusively based on dental features, which, contrary to certain postcranial characters, are not unique to them. The authors report the oldest near-complete skeleton of a fossil flying squirrel (*Miopetaurista neogrivensis*) that displays the gliding-related diagnostic features shared by extant pteromyins and allows for a recalibration of the time of origin and diversification of the group. This study provides a case study to show how fossils can be combined with molecular data to provide better time-calibrated phylogenetic trees, and shows that flying squirrels have experienced few morphological changes for almost 12 million years.

Overall, with beautiful illustrations, this study is of high interest for the general readers, as well as for the researchers. We have the following suggestions to improve the manuscript.

1) This study shows something that isn't explicitly concluded: a fossil squirrel that had been assigned to Pteromyinae on the basis of its dental characters, is now proven to be so on the basis of Pteromyine wrist characters. So while dental characters by themselves might not be completely reliable, in this taxon at least the dental characters were sufficient for assignment to Pteromyinae. This still leaves us unsure of the remainder of fossil "flying squirrel" taxa. This puts some of the discussions under the headings "radiation and biogeography" and "Geographic range…" on uncertain footing, which one reviewer supposed accounts for the term "flying squirrel" often being used with quotations. In statements such as "five species of "flying squirrels" co-occur in a single site" the quotes around "flying squirrels" carry a heavy load of assumption. With the present state of knowledge, that it the best that can be done-the reviewer might just suggest that the authors make it clear in these instances that conclusions depend of the correctness of the subfamily assignment.

In the Introduction it is stated that Emry and Korth (2007) tentatively assigned *Hesperopetes thoringtoni* to a flying squirrel lineage based on its dental characters. In fact, Emry and Korth assigned it to "Subfamily uncertain" after discussing such studies as Mercer and Roth, 2003 and Thorington et al., 2002, and concluding that these "recent morphologic and molecular studies would seem to make it unlikely that the new fossils [*Hesperopetes*] described here represent Pteromyinae." The name *Hesperopetes* suggests similarity to *Oligopetes*, but doesn't necessarily indicate that either genus is in a "flying squirrel lineage."

2) We are less certain how this fossil "casts new light on the origin of the group". The claims that the authors make that "it is highly questionable that 'flying squirrels' older than ca. 31 Ma" appear unfounded, given that their own "total evidence 95% highest posterior density (HPD) interval for the pteromyin divergence is very broad (36.5-24.9 Ma)". The argument seems to be based on node dating analysis estimates, which the authors even state is an alternative method for their analysis. This argument in particular needs to be reconsidered. Most of the Discussion section reads like a review on climate and biogeography of flying squirrels. It is unclear if this is newly compiled information based on new analyses, but it doesn't appear so given the broad agreement with previous studies. If this is indeed the case, it should be clearly stated in the Abstract and Introduction. While the contribution of this fossil is certainly highly valuable, the stated claims that this discovery changes current understanding of flying squirrel evolution seem unfounded.

3) There are many editorial or stylistic errors to be revised. Strangely, we cannot find many cited references in the section "references". It looks that the authors have pasted the section "references" from other papers. Before the re-submission, the authors need double-check the cited references in the main text and the supplementary text.

Regarding the title, how about "oldest near-complete skeleton of.……"? In many cases, we still consider the teeth as a part of skeleton (dermal skeleton, or exoskeleton).

---

## [Author Response]

[…] Overall, with beautiful illustrations, this study is of high interest for the general readers, as well as for the researchers. We have the following suggestions to improve the manuscript.1) This study shows something that isn't explicitly concluded: a fossil squirrel that had been assigned to Pteromyinae on the basis of its dental characters, is now proven to be so on the basis of Pteromyine wrist characters. So while dental characters by themselves might not be completely reliable, in this taxon at least the dental characters were sufficient for assignment to Pteromyinae. This still leaves us unsure of the remainder of fossil "flying squirrel" taxa. This puts some of the discussions under the headings "radiation and biogeography" and "Geographic range…" on uncertain footing, which one reviewer supposed accounts for the term "flying squirrel" often being used with quotations. In statements such as "five species of "flying squirrels" co-occur in a single site" the quotes around "flying squirrels" carry a heavy load of assumption. With the present state of knowledge, that it the best that can be done-the reviewer might just suggest that the authors make it clear in these instances that conclusions depend of the correctness of the subfamily assignment.

We do not entirely coincide with the reviewers in one of their points. *Miopetaurista neogrivensis* was identified as a pteromyin based on dental morphology and the same goes for all extinct purported flying squirrels. However, later research (see Thorington et al., 2005) showed that this assignment, as well as that of all other extinct ‘flying’ squirrels, is unsupported because dental characters used to identify flying squirrels are not unique to them and occur in other squirrel groups. In this work we describe the postcranial anatomy of this taxon and confirm that it belongs to the pteromyins, but this does not validate the unfounded previous ascription to this group solely based on dental morphology.

Regarding the second point raised by the reviewers, quotation marks for the term ‘flying’ squirrels were intended to remark the doubtful attribution of extinct taxa to this group. We have added a new paragraph in the beginning of the “radiation and geography…” section clearly setting this point for the remainder of the discussion: “Except for the partial skeleton of *Miopetaurista neogrivensis* described here, no diagnostic postcranial material has been recovered for any other extinct ‘flying’ squirrel, so their assignment to the pteromyins is doubtful. Here we discuss the fossil record of purported pteromyins in the light of our phylogenetic results, but in these instances our conclusions depend of the correctness of the tribe assignment.”

In the Introduction it is stated that Emry and Korth (2007) tentatively assigned Hesperopetes thoringtoni to a flying squirrel lineage based on its dental characters. In fact, Emry and Korth assigned it to "Subfamily uncertain" after discussing such studies as Mercer and Roth, 2003 and Thorington et al., 2002, and concluding that these "recent morphologic and molecular studies would seem to make it unlikely that the new fossils [Hesperopetes] described here represent Pteromyinae." The name Hesperopetes suggests similarity to Oligopetes, but doesn't necessarily indicate that either genus is in a "flying squirrel lineage."

Emry and Korth (2007) state in the Abstract and elsewhere of their work that “*Hesperopetes* may also represent the oldest occurrence of a lineage leading to pteromyine sciurids”. We agree with the reviewers that this does not mean that these authors considered *Hesperopetes* a flying squirrel at all as it is implicit in the assignment to the Sciuridae without subfamily or tribe ascription.

Furthermore, as outlined by the reviewers, these authors stated that it is unlikely that *Hesperopetes* represents a pteromyin in the light of molecular results. We admit that this was unclear in our initial manuscript so we have added a couple of sentences in the Introduction: “In the light of molecular results, it was conceded that *Hesperopetes* unlikely represented a pteromyin and was not assigned to any squirrel subfamily (Emry and Korth, 2007). On the other hand, this genus appears to have been closely related to *Oligopetes* (Emry and Korth, 2007), an earliest Oligocene (ca. 34–31 Ma) purported flying squirrel from Europe and Pakistan”.

2) We are less certain how this fossil "casts new light on the origin of the group". The claims that the authors make that "it is highly questionable that 'flying squirrels' older than ca. 31 Ma" appear unfounded, given that their own "total evidence 95% highest posterior density (HPD) interval for the pteromyin divergence is very broad (36.5-24.9 Ma)". The argument seems to be based on node dating analysis estimates, which the authors even state is an alternative method for their analysis. This argument in particular needs to be reconsidered. Most of the Discussion section reads like a review on climate and biogeography of flying squirrels. It is unclear if this is newly compiled information based on new analyses, but it doesn't appear so given the broad agreement with previous studies. If this is indeed the case, it should be clearly stated in the Abstract and Introduction. While the contribution of this fossil is certainly highly valuable, the stated claims that this discovery changes current understanding of flying squirrel evolution seem unfounded.

The divergence date estimates for pteromyins provided by our total evidence and node dating analyses do not entirely agree (36.5–24.9 Ma vs. 30.6–17.4 Ma). However, although node dating provides younger estimates for this event the estimates using both independent methods overlap for a time range covering most of the late Oligocene, between 30.6 and 24.9 Ma. Therefore, we considered and still consider this range as the most likely time of divergence between flying and tree squirrels because it is supported by both methods. Notwithstanding, we concede that this should not completely rule out the older estimates provided by total evidence analysis which indeed agree with the age of the oldest alleged ‘flying’ squirrels *Hesperopetes, Oligopetes* and the earliest records of *Sciurion*. In accordance we have reconsidered this argument and decided to tone it down in the current version, clearly stating that these older dates are also supported by one of our methodological approaches but not the other one. This has introduced important modifications in the Discussion (section “Divergence date between flying and tree squirrels”) as well as in the conclusions and Abstract of this work. In addition, the title of the manuscript has been changed from “…casts new light on the origin of the group” to “…casts new light on the phylogeny of the group”. Our results offer a new wealth of information regarding the origin and diversification of flying squirrels, yet they do not close the debate on the controversial assignment to this group of the aforementioned extinct genera. The recovery of diagnostic postcranial material from these earlier forms would likely solve this question.

Regarding the second point, the data on the fossil record of ‘flying’ squirrels have been compiled for this work and include a significant amount of information that had not been considered in recent reviews (see for example Lu et al., 2013). In this section we discuss their fossil record in the light of our phylogenetic results and in the context Neogene and Quaternary climatic evolution. As suggested by the reviewer, this has been clearly stated at the end of the revised Introduction, but it has not been included in the Abstract because of lack of space.

3) There are many editorial or stylistic errors to be revised. Strangely, we cannot find many cited references in the section "references". It looks that the authors have pasted the section "references" from other papers. Before the re-submission, the authors need double-check the cited references in the main text and the supplementary text.Regarding the title, how about "oldest near-complete skeleton of.……"? In many cases, we still consider the teeth as a part of skeleton (dermal skeleton, or exoskeleton).

Errors with the references apparently derived from moving part of the main text to a ‘supplementary text’ file in the earlier version. This supplementary text is now split into several appendices to the main text and included within the article with a shared reference list. We have double checked that all the references are included in that new reference list.

Regarding the title, we acknowledge that teeth are part of the skeleton, but we do not find that it is confusing at all for most paleontologists. Anyway, if the editor finds that changing the title is required we would prefer using “partial” rather than “near-complete”.